# The PavMYB.C2-UFGT module contributes to fruit coloration via modulating anthocyanin biosynthesis in sweet cherry

Yangang Pei[1,2☯], Wanjia Tang[1☯], Yidi Huang[1], Hongfen Li[1], Xiaowei Liu[3], Hongxu Chen[1], Runmei He[1], Wenyi Niu[1], Quanyan Du[1], Yizhe Chu[1], Heng Deng[4], Mingchun Liu[2]*, Ronggao Gong[1]*

1 College of Horticulture, Sichuan Agricultural University, Chengdu, Sichuan, China, 2 Key Laboratory of Bio-Resource and Eco-Environment of Ministry of Education, College of Life Sciences, Sichuan University, Chengdu, Sichuan, China, 3 College of Agriculture and Horticulture, Chengdu Agricultural College, Chengdu, Sichuan, China, 4 School of Life Science and Engineering, Southwest University of Science and Technology, Mianyang, Sichuan, China

☯ These authors contributed equally to this work.
* rggong@sicau.edu.cn (RG); mcliu@scu.edu.cn (ML)

## Abstract

Anthocyanins, vital secondary metabolites responsible for fruit coloration and health benefits, yet the genetic mechanisms regulating anthocyanin biosynthesis in fruits remain incompletely understood. In this study, we conducted a metabolomic analysis that revealed both the total anthocyanin content and the relative abundance of individual anthocyanin species are critical contributors of the color variation observed between yellow- and dark red-fruited cultivars. Integrating transcriptomic data with metabolic profiles, we identified a gene module central to anthocyanin biosynthesis, with PavMYB.C2 emerging as a key transcriptional activator. Functional validation through overexpression and silencing of *PavMYB.C2* in cherry fruit confirmed its essential role in regulating both total anthocyanin and cyanidin-3-glucoside (Cy3G) levels. Furthermore, PavMYB.C2 upregulates transcription of the anthocyanin biosynthetic gene *UFGT* via its serine (S) 68 residue within the MYB domain, leading to enhanced Cy3G accumulation. These findings highlight the PavMYB.C2-UFGT regulatory module as a critical determinant of fruit coloration, offering potential avenues for improving fruit quality through genetic manipulation.

## Author summary

For centuries, the vibrant colors of fruits have intrigued humans, yet the genetic mechanisms underlying these hues remain incompletely understood. Why do some fruits develop a deep red color while others remain yellow? How do plants regulate pigment production during fruit ripening? In this study, we employed a combination of chemical and genetic analyses to investigate the color variation

**Data availability statement:** The raw omics data reported in this paper have been deposited in the National Genomics Data Center (NGDC), China National Center for Bioinformation under the accession numbers CRA021570 and OMIX009876.

**Funding:** This work was supported by the Natural Science Foundation of Sichuan Province, China (grant no. 2024NSFSC0324 and 2025ZNSFSC1100) to RG and YP, the National Key Research and Development Project (grant no. 2017YFC0505104) to RG, the Discipline Construction Dual Support Programme of Sichuan Agricultural University (grant no. 2024ZYTS020) to YP. The funders had no role in study design, data collection and analysis, decision to publish, or preparation of the manuscript.

**Competing interests:** The authors have declared that no competing interests exist.

between yellow and dark-red cherries. Our results revealed that color differences are not solely attributable to overall anthocyanin levels but also to specific pigments, particularly cyanidin-3-glucoside (Cy3G). A key player identified in this process was the gene *PavMYB.C2*, which functions as a genetic "switch" controlling anthocyanin synthesis. By manipulating the expression of *PavMYB.C2* in cherry fruits, we demonstrated its role in both general pigment production and the specific enhancement of Cy3G levels. Furthermore, we elucidated a critical molecular mechanism in which PavMYB.C2 activates the expression of *UFGT*, a key gene in the final step of Cy3G biosynthesis. These findings provide valuable insights into the molecular basis of fruit color variation.

## Introduction

Anthocyanins, a subclass of water-soluble flavonoid pigments, are increasingly recognized for their health-promoting properties [1–3]. They exhibit potent antioxidant capabilities by scavenging reactive oxygen species (ROS), providing protection against chronic diseases like cancer and cardiovascular disorders in humans [2,4]. Anthocyanins are essential for the pigmentation of various plant tissues, such as flowers, fruits, and leaves, alongside their health benefits [5,6]. Among these, fruits serve as one of the primary food sources for humans, naturally becoming a medium for anthocyanin intake. Within fruit cells, anthocyanins are predominantly stored in vacuoles, where they impart a rich array of fruit colors, ranging from red to purple and blue [7–9]. These multifaceted roles of anthocyanins underscores their significance in both plant physiology and human health. Therefore, exploring strategies to enhance anthocyanin content through genetic, environmental, and biochemical approaches—holds promise not only for enhancing the appearance quality of fruits but also for augmenting their nutritional value.

As a specific end product of the flavonoid biosynthesis pathway, anthocyanins frequently combine with various monosaccharides, facilitated by specific enzymes, to form stable anthocyanins [10]. Anthocyanins biosynthesis initiates with phenylalanine, which undergoes enzymatic transformations involving phenylalanine ammonia lyase (PAL), chalcone synthase (CHS), chalcone isomerase (CHI), flavanone 3-hydroxylase (F3H), flavonoid 3′-hydroxylase (F3′H), dihydroflavonol reductase (DFR), anthocyanin synthase (ANS), and anthocyanin 3-O-glucotransferase (UFGT) [11–14]. The enzymes CHS, CHI, F3H, and F3′H are essential in the early stages of anthocyanin synthesis, whereas DFR, ANS, and UFGT are vital in the later stages. For instance, the expression level of *DFR* can influence plant coloration, with *DFR* levels being highest in organs that accumulate significant amounts of anthocyanins [15]. ANS catalyzes the conversion of proanthocyanidins to colored anthocyanins, and the deletion of the *ANS* gene results in a reduced anthocyanin content [16]. UFGT is the final enzyme in the anthocyanin synthesis pathway, facilitating the conversion of unstable anthocyanins into stable forms through glycosylation [17]. Numerous anthocyanins have been identified in nature, primarily including paeonidin, cyanidin,

pelargonidin, delphinidin, petunidin, and malvidin [11]. Notably, the dominant anthocyanin components vary significantly among various fruit species. For example, grapes (*Vitis vinifera*) are rich in delphinidin and cyanidin [18], while blueberries (*Vaccinium spp.*) primarily contain delphinidin and petunidin [19], and purple tomatoes (*Solanum chilense*) exhibit high concentrations of delphinidin and petunidin [20]. These observations indicate distinct mechanisms of anthocyanin synthesis across various fruit species.

Transcription factors are crucial proteins that modulate gene expression by acting as either activators or repressors. They achieve this by binding to *cis*-acting regulatory elements located within the promoter regions of target genes [21]. Recent research has uncovered a variety of transcription factors that participate in the regulation of anthocyanin biosynthesis, including members of the MYB (Myeloblastosis), bHLH (basic Helix-Loop-Helix), WRKY (WRKY DNA-binding protein), NAC (NAM, ATAF1/2, CUC2), bZIP (Basic Leucine Zipper), and ERF (Ethylene Response Factor) families [22–25]. Among these, the MYB family has garnered significant attention due to its central role in regulating anthocyanin biosynthesis. They regulate the expression of target genes involved in this biosynthetic pathway by specifically binding to cis-regulatory elements such as E-boxes and G-boxes [25]. For instance, in *Arabidopsis thaliana*, MYB factors like AtMYB75/PAP1, AtMYB90/PAP2, and AtMYB113/114 are well-documented as positive regulators of anthocyanin biosynthesis [26–28]. Similarly, in the Rosaceae plants, MYB1 and MYB10 have been shown to enhance the expression of key genes such as *DFR*, *ANS*, and *UFGT*, thereby regulating the biosynthesis of anthocyanins [29–31].

A comprehensive understanding of the metabolic pathways and regulatory networks controlling fruit flavor and nutritional quality is essential for effective trait manipulation [32]. The integration of multi-omics strategies has illuminated how specific genes and transcription factors regulate flavor-associated metabolites during fruit development and ripening [33,34]. Recent transcriptomic and metabolomic studies have enhanced our comprehension of the mechanisms and regulation of anthocyanin synthesis in various fruit species. For instance, a comprehensive analysis of the anthocyanin biosynthesis pathway in tomato identified the transcription factor SlAN2-like acts as a key regulator, activating the expression of both structural and regulatory genes involved in anthocyanin accumulation in response to light [35]. In *Gardenia jasminoides*, the integration of transcriptomic and metabolomic approaches has pinpointed critical pathways influencing color formation, revealing a wealth of differentially expressed genes linked to photosynthesis and pigment biosynthesis [36]. This integrative approach has also been applied to cashew apples, where distinct anthocyanin compounds were identified, and their biosynthetic genes were shown to correlate with the observed color variations across different cultivars [37]. These findings emphasize the integration of transcriptomic and metabolomic analyses offers a powerful framework for dissecting the complex regulatory mechanisms of anthocyanin biosynthesis in fruits.

Sweet cherry (*Prunus avium* L.), part of the Rosaceae family's *Prunus* genus, is popular among consumers for its anthocyanin-rich fruit [38]. Various cultivars of sweet cherry display a diverse spectrum of colors, ranging from yellow to deep red, which are intricately associated with their anthocyanin content [39,40]. Previous studies have identified several families of transcription factors involved in the regulation of anthocyanin biosynthesis in sweet cherry, including PavNAC56, PaMADS7, and PavBBX6/9, which modulate anthocyanin accumulation during the abscisic acid (ABA)-mediated fruit ripening process [41–43]. Notably, one study constructed a population of varieties with distinct peel colors and localized a key gene encoding the MYB transcription factor PavMYB10.1 [39]. This gene is believed to form a putative MBW-activating complex with PavbHLH and PavWD40, specifically regulating anthocyanin biosynthesis [39]. This underscores the essential function of MYB transcription factors in regulating anthocyanin synthesis in sweet cherry. However, few studies to date have integrated anthocyanin metabolism with transcriptomic data to explore the regulatory mechanisms underlying color formation in sweet cherry.

In this study, we investigated two sweet cherry cultivars, 'Binghu' and 'Isabella', which exhibit uniform ripening progress but distinct color differences. We generated the metabolomes and transcriptomes of sweet cherries at five fruit ripening stages using a multi-omics approach, clarifying the effects of variations in anthocyanin content and species on the formation of sweet cherry color. A regulatory network for anthocyanin biosynthesis was constructed through comprehensive

bioinformatics analysis, leading to the identification of a gene encoding a MYB transcription factor, *PavMYB.C2*. Genetic and molecular biology experiments collectively demonstrated that PavMYB.C2 acts as a key activator in regulating anthocyanin biosynthesis. This study provides novel insights into the regulation of sweet cherry color formation and anthocyanin metabolism, establishing a basis for enhancing sweet cherry fruit quality.

## Results

### Differential accumulation of anthocyanins between two cultivars exhibiting variations in fruit coloration

Sweet cherries display diverse fruit skin colors, such as yellow, orange, blush, and dark red [44]. To explore the mechanisms underlying the formation of fruit color in sweet cherries, we selected two cultivars, 'Binghu' and 'Isabella', which display marked differences in fruit coloration (Fig 1A). The color changes were tracked through five stages, from the breaker stage (Br) to 20 days post-breaker (Br20), to observe the transition from color onset to full ripening (Fig 1A). Our

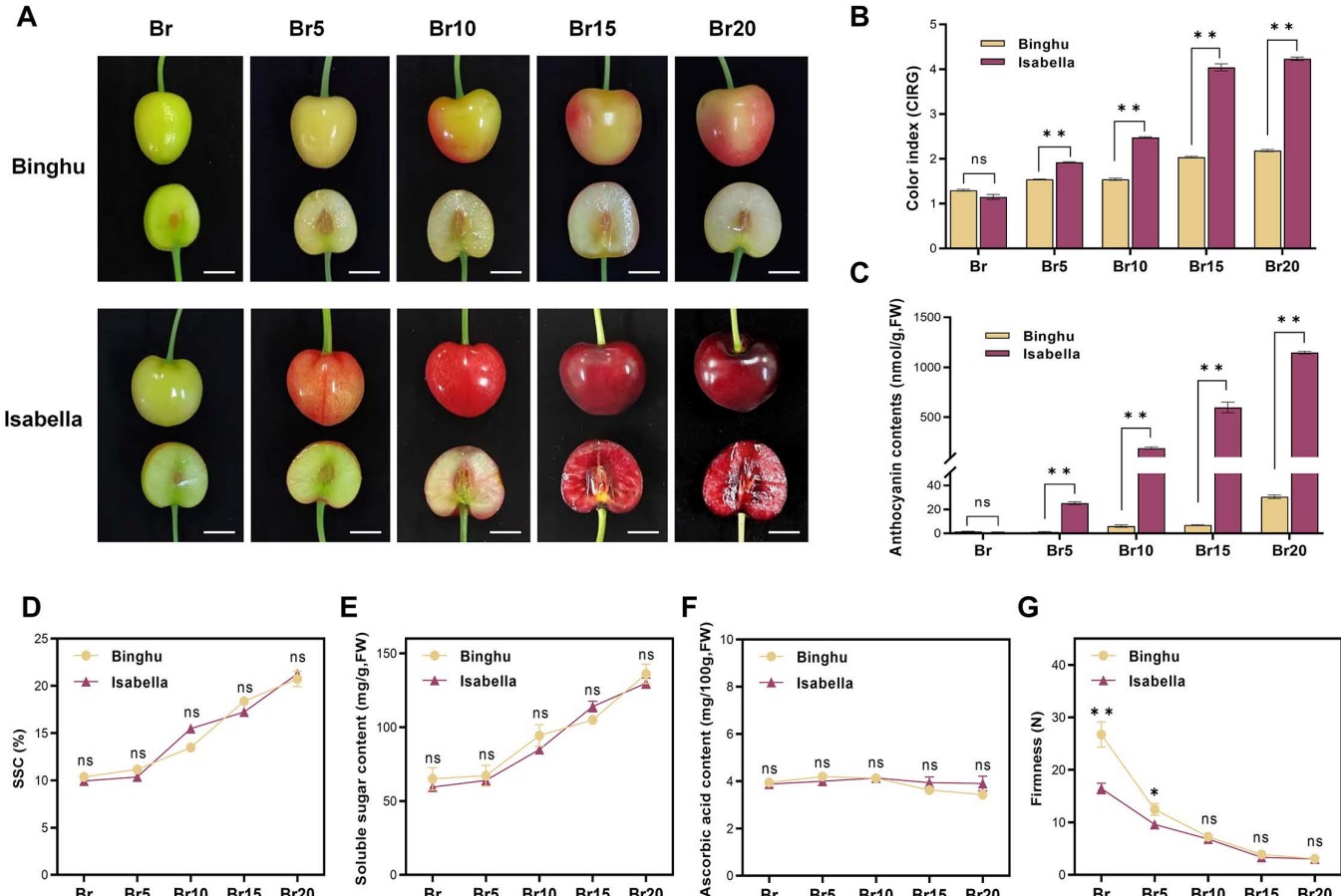

**Fig 1. Differential accumulation of anthocyanins between two cultivars with distinct fruit coloration. (A)** Phenotypes comparison of 'Binghu' and 'Isabella' sweet cherry fruits at different ripening stages. 'Binghu' and 'Isabella' are two commonly cultivated sweet cherry cultivars. Ripening stages include the breaker stage (Br) and 5-20 days post-breaker stage (Br5-20). Scale bar = 1 cm. **(B)** Color index of red grape (CIRG) values for the fruits at each ripening stage. **(C)** Anthocyanin content in the fruits at each ripening stage. **(D-G)** Analysis of fruit ripening-related phenotypes. **(D)** Soluble solid content (SSC), **(E)** Soluble sugar content, **(F)** Ascorbic acid content, and **(G)** Fruit firmness at each ripening stage. Data are presented as means ± standard deviation (SD) (n = 5). Statistical significance was determined by Student's *t*-tests; asterisks denote significant differences compared to 'Binghu' at $p < 0.05$ (*), $p < 0.01$ (**); 'ns' indicates no significant difference compared to the control.

observations indicated that 'Binghu' fruits transitioned from green to yellow, with light red hues emerging as the fruit ripened. In contrast, 'Isabella' exhibited a more pronounced color change, culminating in a dark red hue at full ripeness (Fig 1A). Fruit color index measurements confirmed significant differences in fruit color between the two cultivars, and these differences became more pronounced as ripening advanced (Fig 1B). Anthocyanin content analysis also showed that 'Isabella' fruits contained significantly higher anthocyanin levels than 'Binghu', particularly at the later stages of ripening (Fig 1C). To determine whether this variation in color development and anthocyanin accumulation was attributable to broader ripening-related changes, we assessed additional ripening traits. Both cultivars exhibited an increase in soluble sugars and soluble solids (Fig 1D and 1E), a slight decrease in organic acid content (Fig 1F), and a marked reduction in fruit firmness as ripening progressed (Fig 1G). However, the ripening traits showed no significant differences between the two cultivars, particularly during the later ripening stages (Fig 1D–1G). These findings suggest that the observed differences in fruit color are likely attributable to cultivar-specific variations in anthocyanin accumulation, rather than differences in overall ripeness.

## Metabolic profiling of anthocyanin-associated compounds between two cultivars during fruit ripening

To further elucidate the metabolic mechanisms underlying color formation in sweet cherries, we collected fruit samples at five ripening stages from two cultivars described in Fig 1. Liquid chromatography tandem mass spectrometry (LC-MS/MS)-based metabolic profiling was utilized to monitor the dynamic changes in anthocyanin levels throughout these stages. A total of 31 anthocyanins and six proanthocyanidins were identified, with the anthocyanins further classified into five distinct categories: cyanidins, pelargonidins, peonidins, petunidins, and delphinidins (Fig 2A; S1 Data). Notably, anthocyanin accumulation was significantly higher in Isabella cherries, with concentrations increasing progressively during ripening, particularly during the mid- to late stages (Br10, Br15, and Br20) (Fig 2A). In contrast, proanthocyanidin content did not differ markedly between cultivars and was predominantly concentrated in the early ripening stages (Br and Br5) (Fig 2A). These findings suggest that anthocyanins are key contributors to fruit color development in sweet cherries. Principal component analysis (PCA) revealed that while metabolic differences between cultivars were minimal at the onset of color change, they became more pronounced as ripening advanced (Fig 2B). Consequently, hierarchical clustering did not group samples from the two cultivars at the same ripening stage together (Fig 2B). Overall, these results indicate that the two cherry cultivars exhibit distinct anthocyanin metabolic profiles during fruit ripening.

To identify the key anthocyanin fractions responsible for sweet cherry color formation, we analyzed differential metabolites (DMs) between the 'Binghu' and 'Isabella' cultivars at each ripening stage. Differential metabolites were defined as those with p-values < 0.05 and VIP > 1, with all data provided in S1 Data. Given the substantial accumulation of anthocyanins during the mid- to late ripening stages (Fig 2A), we focused on stages Br10 (17 DMs), Br15 (19 DMs), and Br20 (20 DMs). Venn diagrams revealed overlap among the differential metabolites, with 15 DMs consistently differentially accumulating across all three stages, indicating that most anthocyanin species exhibit differential accumulation between the two cultivars (Fig 2C). Among these, four anthocyanins—Cyanidin-3-O-rutinoside (Cy3R), Cyanidin-3-O-glucoside (Cy3G), Peonidin-3-O-rutinoside (Pn3R), and Pelargonidin-3-O-rutinoside (Pg3R)—accounted for over 90% of the total anthocyanin content in 'Isabella', with a marked increase in their levels during ripening (Figs 2D and S1A). These findings suggest that the significant differences in color between the two cultivars are primarily due to variations in anthocyanin content, especially the accumulation of these four key anthocyanidins. We also examined the relative abundance of individual anthocyanin species across the five ripening stages. In 'Binghu', the relative proportions of anthocyanins remained relatively stable, with Cy3R being the dominant compound at all stages (S1A Fig). In contrast, 'Isabella' exhibited a shift in the relative composition over ripening, with a notable increase in the proportion of Cy3G, alongside a rise in Pn3R and Pg3R (S1A Fig). Furthermore, color index analysis of standard solutions of these four anthocyanins revealed that Cy3G produced a deeper red hue compared to the others (S1B Fig). Collectively, these results highlight that the differences in fruit color between the two cherry cultivars are not only driven by total anthocyanin content but also by the specific anthocyanin profile and the dynamic changes in their relative abundance during ripening.

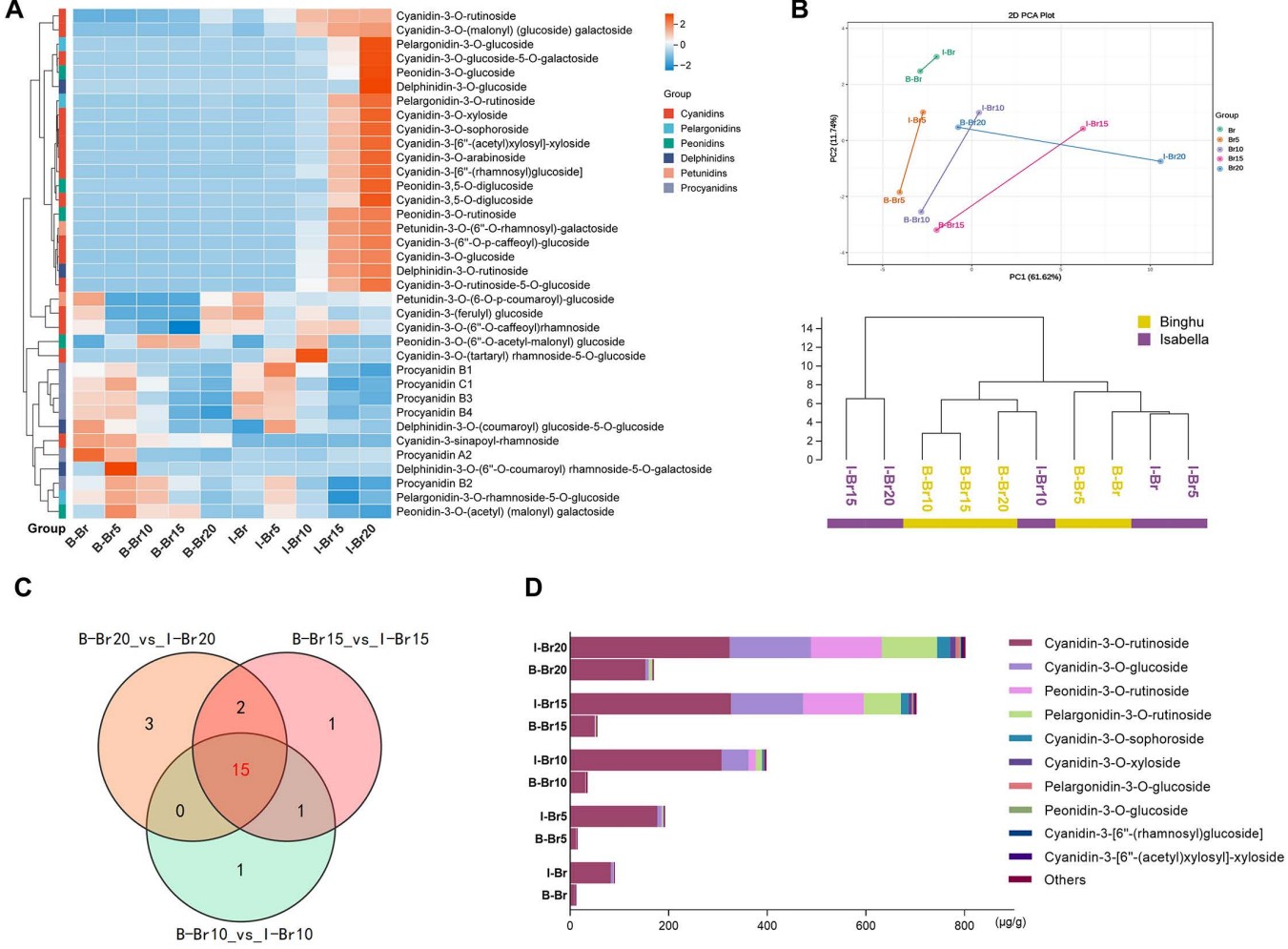

**Fig 2. Anthocyanin-related metabolome analysis in two sweet cherry cultivars across ripening stages. (A)** Overview of anthocyanin-related metabolome across five ripening stages of 'Binghu' (abbreviated as B) and 'Isabella' (abbreviated as I) sweet cherry fruits. **(B)** Principal component analysis (PCA) and cluster dendrogram of metabolome data from the five ripening stages of both cultivars. The distance between two data points represents the degree of similarity between different groups. **(C)** Venn diagram showing the overlap of differential metabolites (DMs) between 'Binghu' and 'Isabella' fruits at Br10, Br15 and Br20 stages. Red letters indicate the number of DMs common to all three stages. **(D)** Statistics of the class categories of metabolites at five ripening stages. Different classes of metabolites are identified by different colours.

## Genetic basis of anthocyanin dynamics in two cultivars

RNA-Seq analysis was conducted across five distinct ripening stages of sweet cherries to investigate the genetic basis of anthocyanin-related metabolism. A total of 96 Gb of raw data were generated, resulting in the annotation of 28,007 genes (S2 Data). Based on expression profiles at different ripening stages, genes were classified into three groups: Group I, with high expression at the late ripening stage; Group II, highly expressed at the early ripening stage; and Group III, with expression patterns that varied across stages (Fig 3A). In contrast to the metabolite profiles, PCA analysis revealed a high similarity between samples obtained from two cultivars at corresponding ripening stages (Fig 3B). Clustering analysis further confirmed that samples from both cultivars grouped together at each ripening stage (Fig 3B), suggesting that, despite their distinct color characteristics, the ripening processes of 'Binghu' and 'Isabella' are genetically similar.

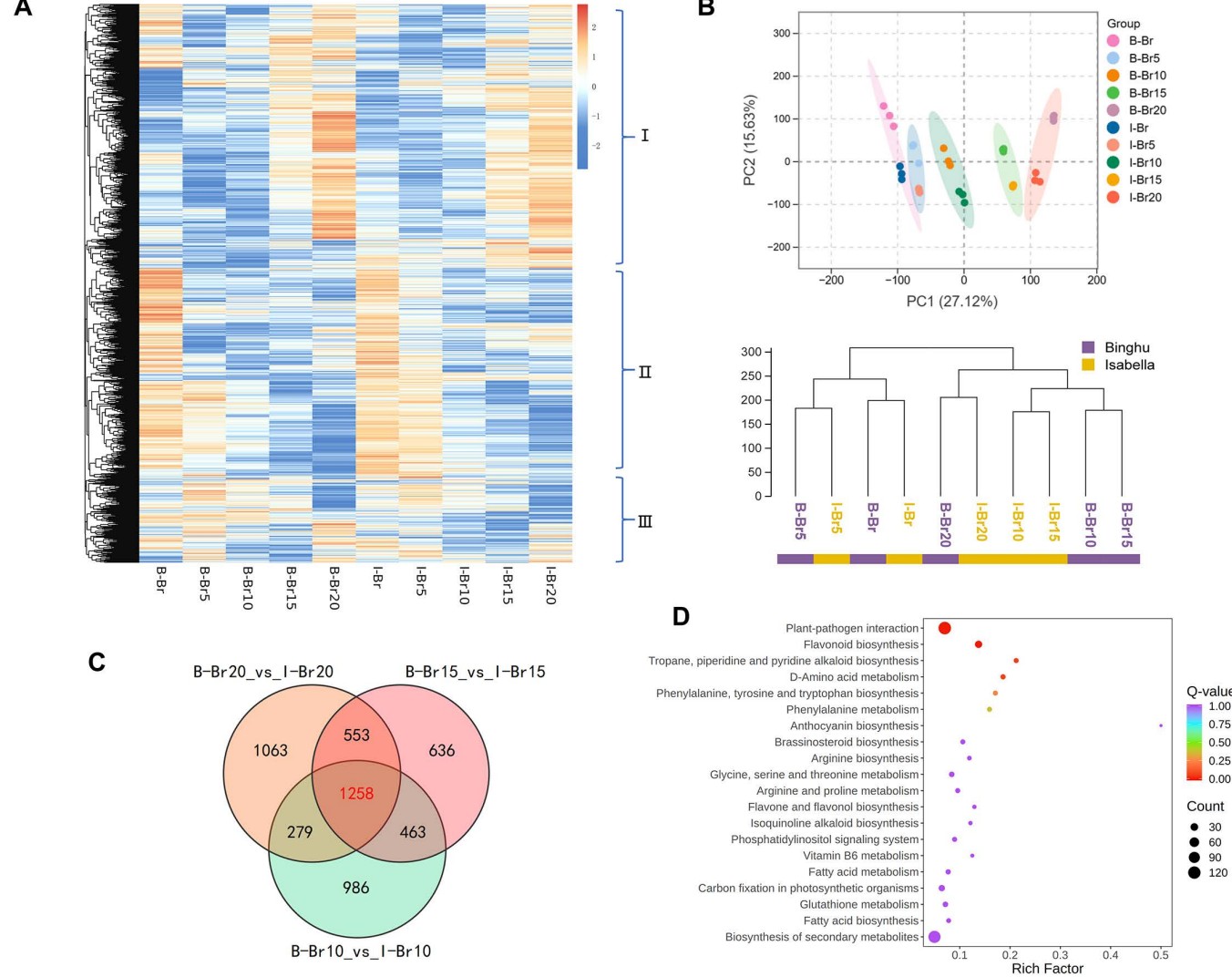

**Fig 3. Transcriptome analysis in two sweet cherry cultivars across ripening stages. (A)** Transcriptomic data are divided into three clades during fruit ripening. Genes in group I are highly expressed at the late ripening stage; Genes in group II are highly expressed at the early ripening stage; Genes in group III are expressed varied across stages. **(B)** PCA and cluster dendrogram of transcriptome data from the five ripening stages of both cultivars. The oval circle indicates that it is a 95% confidence ellipse. **(C)** Venn diagram showing the overlap of differentially expressed genes (DEGs) between 'Binghu' and 'Isabella' fruits at Br10, Br15 and Br20 stages. Red letters indicate the number of DEGs common to all three stages. **(D)** KEGG enrichment bubble diagram to show the first 20 pathways with the smallest q values, pathway shown as the ordinate and rich factor shown as the abscissa.

Although the overall transcriptional profiles of the two cultivars were similar, we hypothesized that key genes involved in color formation might still show differential expression. To investigate this, gene expression was analyzed for each cultivar at the Br10, Br15, and Br20 stages, identifying differentially expressed genes (DEGs) with a fold change greater than 1.5 and a $P$-value less than 0.05. The analysis revealed 2,986, 2,910, and 3,153 DEGs at these respective stages (S2A–S2C Fig). In 'Isabella' fruits, compared to 'Binghu', 1,815, 1,160, and 1,951 genes were upregulated, while 1,171, 1,750, and 1,202 genes were downregulated at the same stages (S2A–S2C Fig). KEGG pathway analysis of the DEGs revealed significant enrichment in key metabolic pathways, including flavonoid biosynthesis, biosynthesis of secondary metabolites, amino sugar and nucleotide sugar metabolism, phenylpropanoid biosynthesis, glutathione metabolism, and cysteine and

methionine metabolism (S2D–S2F Fig). Venn diagram analysis further revealed 1,258 DEGs that were consistently differentially expressed across all three stages (Fig 3C; S3 Data). These genes were notably enriched in pathways associated with plant-pathogen interactions, flavonoid biosynthesis, secondary metabolite biosynthesis, and fatty acid biosynthesis (Fig 3D; S4 Data). Notably, flavonoid biosynthesis is the primary pathway for anthocyanin biosynthesis, suggesting that the identified DEGs likely play significant roles in the accumulation of anthocyanins in sweet cherries.

### Identification of genetic modules involved in anthocyanin biosynthesis by combined transcriptomics and metabolomics analysis

We then performed Weighted Gene Co-expression Network Analysis (WGCNA) on combined transcriptomic and metabolomic data, including 15 differentially expressed metabolites (DMs) and 1,258 differentially expressed genes (DEGs) (Fig 3C), to identify gene modules co-expressed with anthocyanin metabolites. The WGCNA hierarchical clustering tree (Fig 4A) illustrates the relationships between genes, with each branch representing a gene and color bands indicating different co-expression modules. We identified five distinct gene modules based on their similar expression patterns (Fig 4A). Among them, the turquoise and blue modules contain the largest number of genes, accounting for 45.68% and 36.54% of all DEGs, respectively (Fig 4B). The module correlation heatmap highlights reveals a strong association between the blue module and most anthocyanin-related DMs (Fig 4C; S5 Data). The gene significance vs. metabolite significance diagram (Fig 4D–4H) further confirms the blue module's prominence, showing that the genes within this module have high gene significance values for anthocyanin metabolites. This robust gene-metabolite association underscores the genes' role in regulating anthocyanin biosynthesis, indicating that their expression might affect metabolite accumulation.

### Generation of anthocyanin metabolic regulatory networks

In the blue module, we identified 10 genes involved in anthocyanin metabolism, including key biosynthetic genes that encode chalcone synthase (CHS), chalcone isomerase (CHI), flavanone 3-hydroxylase (F3H), flavonoid 3'-hydroxylase (F3'H), dihydroflavonol 4-reductase (DFR), anthocyanidin synthase (ANS), and UDP-glucose:flavonoid 3-O-glucosyltransferase (UFGT) (S5 Data). These genes encode well-known key enzymes of anthocyanin metabolic pathway (Fig 5A). The gene expression profiles showed a strong correlation with the accumulation of the four major anthocyanins, confirming the robustness of the constructed blue module (S5 Data; Fig 5B). Additionally, we identified 18 transcription factors, including members of the ethylene response factor (ERF), MYB, and basic leucine zipper (bZIP) families, which, based on their correlation with both the structural genes and anthocyanin metabolism, establish a complex regulatory network within the blue module (S5 Data; Fig 5B). These transcription factors are probably essential for regulating anthocyanin biosynthesis in sweet cherry fruit.

Meanwhile, to validate the RNA-seq data, we performed RT-qPCR analyses on fruit samples from two sweet cherry cultivars, 'Isabella' and 'Binghu', across five stages of ripening. We selectively measured the expression levels of three structural genes (DFR, ANS, UFGT) involved in anthocyanin synthesis. In 'Isabella', the expression of these genes exhibited a progressive increase during fruit ripening, aligning with the RNA-seq data (Fig 5C–5E; S2 Data). Conversely, 'Binghu' displayed a less pronounced increase in the expression of DFR, ANS and UFGT. (Fig 5C–5E). Notably, the expression levels of these critical genes were markedly elevated in the 'Isabella' compared to the 'Binghu' cultivar, especially during the later stages of ripening (Br10, Br15, and Br20) (Fig 5C–5E). Correspondingly, the enzymatic activities of DFR, ANS, and UFGT in 'Isabella' were substantially higher than those in 'Binghu' (Fig 5F–5H). This observation implies significant disparities in the regulatory mechanisms of anthocyanin biosynthesis between the two cultivars.

### A MYB transcription factor-driven enhancement of anthocyanin accumulation in sweet cherry fruit

In various fruit species, certain members of the MYB transcription factor family have been demonstrated to play crucial roles as regulators in the biosynthesis of anthocyanins [25,28,45–47]. Within our constructed regulatory network for anthocyanin metabolism, we identified a gene encoding a MYB-related transcription factor, LOC110764247, which exhibited

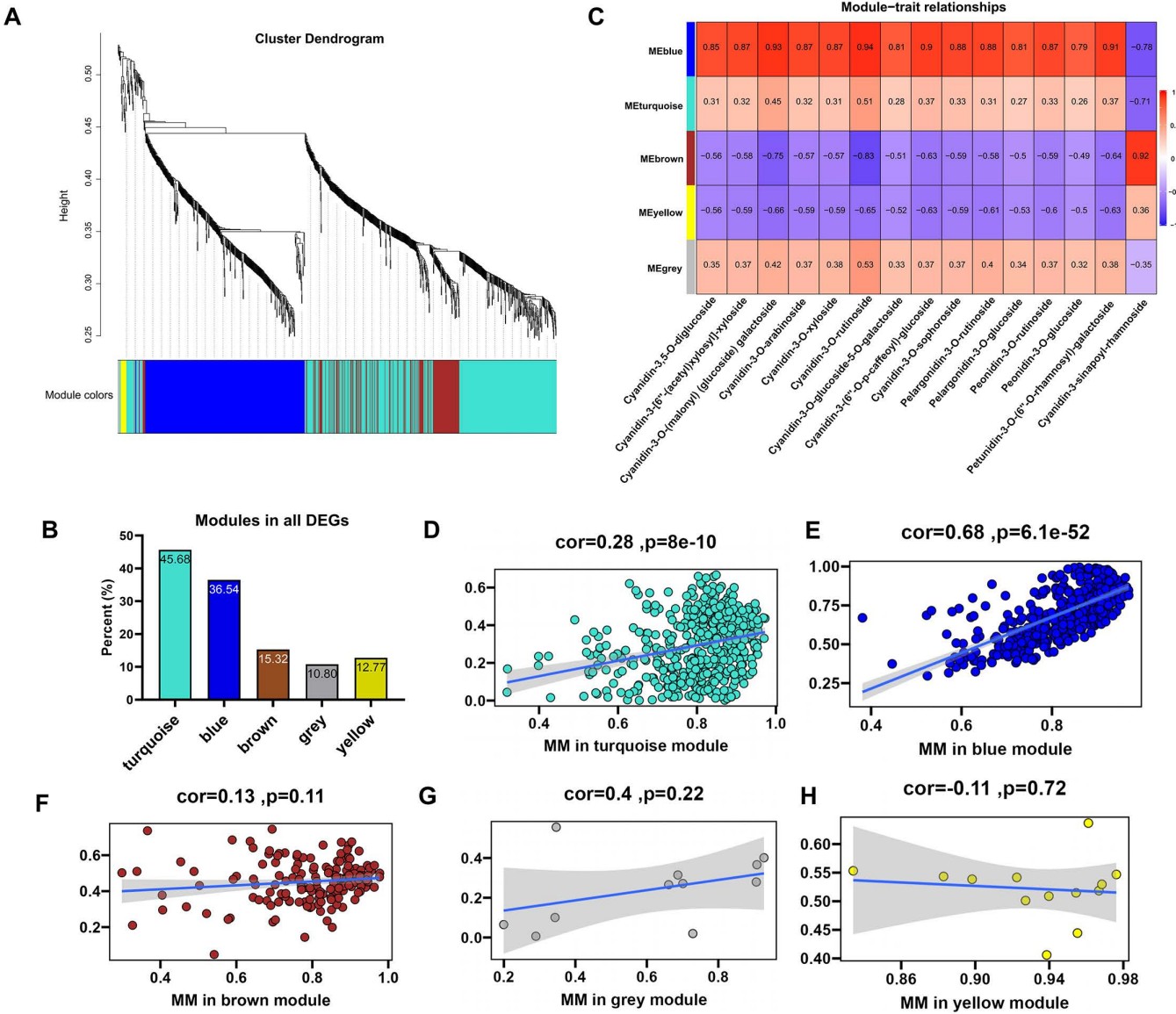

**Fig 4. Identification of genetic modules associated with anthocyanin biosynthesis. (A)** Co-expression modules identified by weighted correlation network analysis (WGCNA) across fruit ripening stages. The main branches of the tree represent five distinct modules, each assigned a unique color to denote its respective group. **(B)** Heat map displaying the correlation between each module and anthocyanin-related compounds. Each column corresponds to a module marked with a different color. Rows represent individual co-expression modules, while columns correspond to specific anthocyanin components. Red indicates a positive correlation, while blue signifies a negative correlation. **(C)** Proportion of differentially expressed genes (DEGs) within each of the identified modules, expressed as a percentage of the total number of DEGs. **(D-H)** Analysis of gene significance and module membership (MM) for genes associated with anthocyanin biosynthesis in the five key modules.

a strong correlation with the accumulation of anthocyanins (Fig 5A). Furthermore, RT-qPCR analysis revealed differential expression of LOC110764247 between the two cultivars, with a notable upregulation during the ripening process in 'Isabella' fruit, suggesting a potential function in anthocyanin biosynthesis (Fig 6A). Phylogenetic analysis using full-length amino acid sequences (Fig 6B), MYB domain sequences (S3A Fig), and full-length coding DNA sequences (CDS) (S3B Fig) collectively demonstrates that LOC110764247 clusters within the CPC (CAPRICE)-like MYB transcription factor

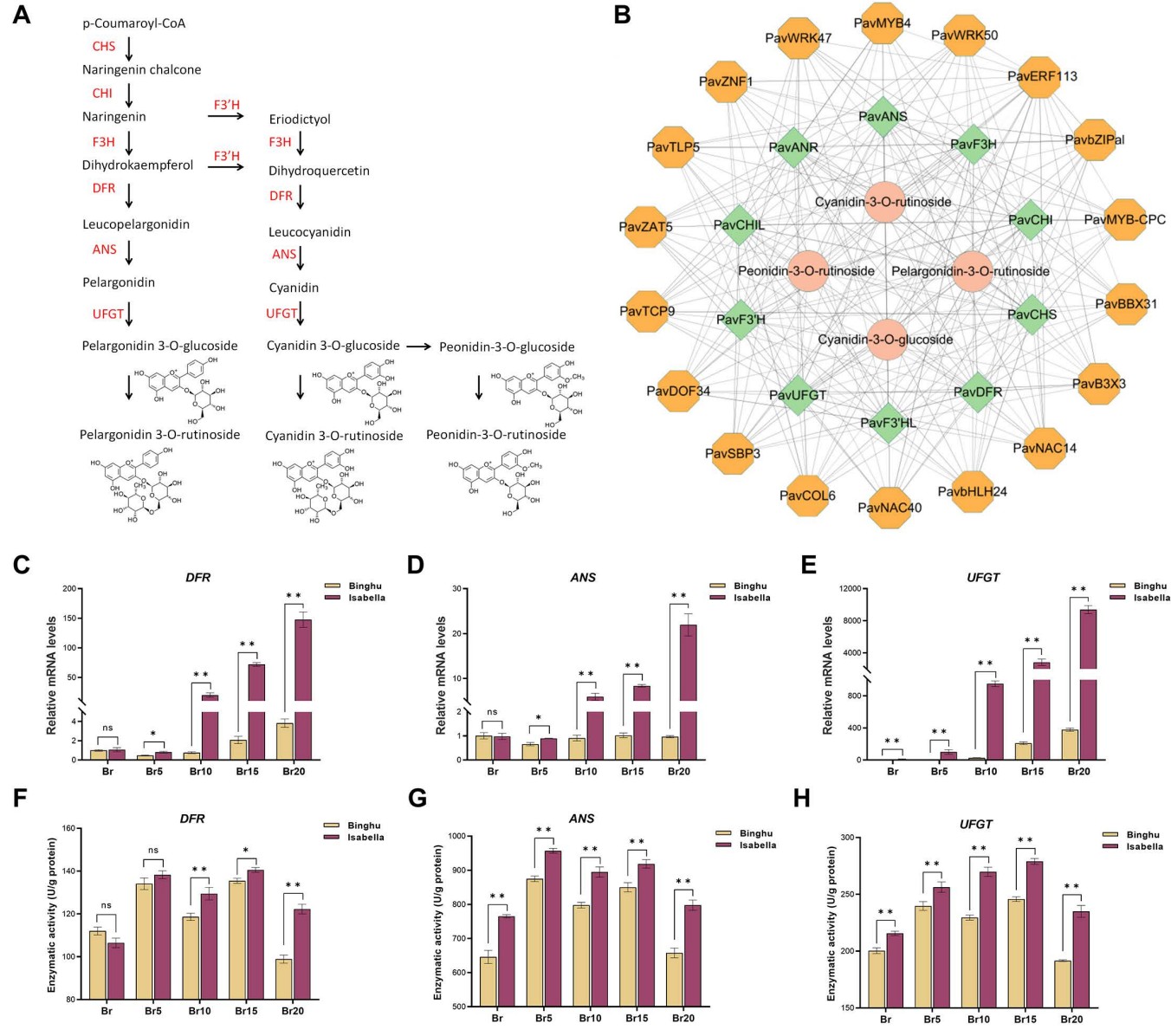

**Fig 5. Construction of anthocyanin metabolic regulatory networks. (A)** Overview of the anthocyanin biosynthetic pathway. Key enzymes involved in anthocyanin biosynthesis are highlighted in red. **(B)** Regulatory network of major anthocyanin metabolites. Pink circles represent anthocyanin compounds, green diamonds indicate structural genes involved in anthocyanin metabolism, and yellow polygons denote transcription factors identified in the blue module, whose expression correlates with both the structural genes and anthocyanin biosynthesis. **(C-E)** Comparative analysis of the relative transcript levels of the structural genes *DFR* (C), *ANS* (D), and *UFGT* (E) in two cultivars across five stages of fruit ripening. **(F-H)** Measurement of enzyme activity for DFR (F), ANS (G), and UFGT (H) in the two cultivars throughout the five ripening stages. Data are presented as means ± standard deviation (SD) (n = 3). Statistical significance was assessed using Student's *t*-tests. Asterisks denote significant differences compared to 'Binghu' at $p < 0.05$ (*), $p < 0.01$ (**); 'ns' indicates no significant difference compared to the control.

clade, yet forms a distinct branch that is divergent from the AtCPC protein in *Arabidopsis*. This distinctive positioning supports its designation as the MYB transcription factor CPC 2 (PavMYB.C2). Subsequently, we examined the subcellular localization of this transcription factor by introducing the recombinant vectors PavMYB.C2-eGFP and free-eGFP into

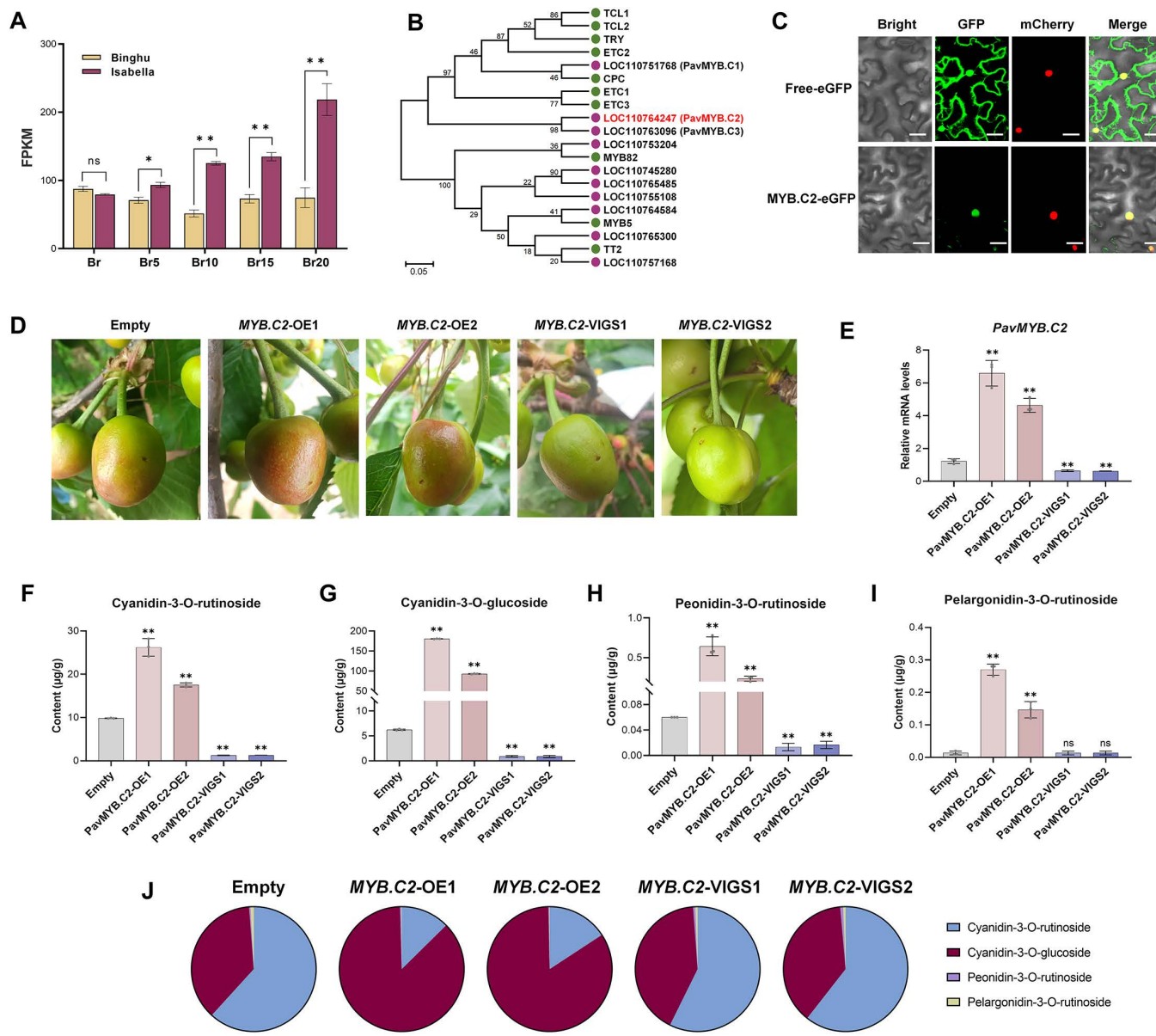

**Fig 6. PavMYB.C2-driven enhancement of anthocyanin accumulation in sweet cherry fruit. (A)** Expression levels of the LOC110764247 (*PavMYB.C2*) gene in two cultivars at five ripening stages, as determined by RNA-seq. **(B)** Phylogenetic analysis of the LOC110764247 (PavMYB.C2) homologs in *Arabidopsis thaliana* and *Prunus avium* was performed using full-length amino acid sequences. The tree was constructed using the Maximum Likelihood (ML) method. **(C)** Subcellular localization of PavMYB.C2. Recombinant vectors (PavMYB.C2-eGFP and free-eGFP) were introduced into *Nicotiana benthamiana* mesophyll cells. Subcellular localization was determined by observing GFP fluorescence under a fluorescence microscope, with co-expression of a nucleus-targeted mCherry marker to delineate nuclear regions. Scale bar = 20 μm. **(D)** Transient genetic transformation in sweet cherry fruit. 'OE' denotes transient overexpression, 'VIGS' refers to transient gene silencing, and 'Empty' indicates the injection of a corresponding empty vector. Images were captured 15 days post-injection. **(E)** Relative transcript levels of *PavMYB.C2* in 'Empty', 'OE', and 'VIGS' fruit 15 days after injection. **(F-I)** Quantification of four major anthocyanins in 'Empty', 'OE', and 'VIGS' fruits 15 days post-injection. F: Cyanidin-3-O-rutinoside (Cy3R); G: Cyanidin-3-O-glucoside (Cy3G); H: Peonidin-3-O-rutinoside (Pn3R); I: Pelargonidin-3-O-rutinoside (Pg3R). **(J)** Relative abundances of various anthocyanins in 'Empty', 'OE', and 'VIGS' fruits 15 days after injection. Data are presented as means ± standard deviation (SD) (n = 3). Statistical significance was assessed using Student's *t*-tests. Asterisks denote significant differences compared to 'Empty' at $p < 0.05$ (*), $p < 0.01$ (**); 'ns' indicates no significant difference compared to the control.

*Nicotiana benthamiana* mesophyll cells (Fig 6C). The results demonstrated that PavMYB.C2-eGFP green fluorescent signals were mainly observed in the nucleus, suggesting PavMYB.C2's nuclear localization, aligning with typical transcription factor behavior (Fig 6C).

To further investigate the role of PavMYB.C2 in anthocyanin metabolism, we constructed fusion vectors for the overexpression and virus-induced gene silencing (VIGS) of *PavMYB.C2* to assess its impact on anthocyanin accumulation in sweet cherry fruit in *vivo*. Agrobacterium strain *GV3101* containing the recombinant plasmids 35S:*MYB.C2* or pTRV2:*MYB.C2* was introduced into fruit at 20 days post-anthesis (DPA) (Fig 6D). The success of the genetic transformation was confirmed by RT-qPCR analysis conducted 15 days post-injection (Fig 6E). Overexpression of *PavMYB.C2* (*MYB.C2*-OE) led to a significant upregulation of *PavMYB.C2* expression (Fig 6E), while silencing via VIGS (*MYB.C2*-VIGS) resulted in a marked downregulation of the gene (Fig 6E). Visual examination revealed that *MYB.C2*-OE fruits exhibited a deeper red coloration compared to controls, while *MYB.C2*-VIGS fruits appeared greener (Fig 6D). This phenotypic variation was accompanied by a significant increase in the levels of the four major anthocyanins—Cy3R, Cy3G, Pn3R, and Pg3R—in the *MYB.C2*-OE fruits (Fig 6F–6I). Conversely, these anthocyanins were significantly reduced in the *MYB.C2*-VIGS fruits (Fig 6F–6I). These findings confirm that PavMYB.C2 directly regulates anthocyanin biosynthesis in sweet cherry. Notably, the relative abundance of these anthocyanins were also altered. In addition to increasing the total anthocyanin content, *PavMYB.C2* overexpression shifted the distribution of individual anthocyanins, particularly enhancing the proportion of Cy3G (Fig 6G and 6J). This suggests that MYB.C2 not only promotes overall anthocyanin accumulation but also influences the specific distribution of anthocyanin types within the fruit. These results provide evidence that PavMYB.C2 plays a central role in regulating both the accumulation and composition of anthocyanins in sweet cherry.

## The expression of anthocyanin synthesis-related genes is regulated by PavMYB.C2

To gain the molecular mechanisms by which PavMYB.C2 regulates anthocyanin biosynthesis in fruit, we performed RNA sequencing (RNA-seq) on *MYB.C2*-OE and control (empty) fruits at 15 days post-injection. PCA revealed clear separation between *MYB.C2*-OE and control samples along principal components 1 and 2 (S4A Fig), indicating substantial differences in gene expression profiles. A expression heatmap highlighted significant alterations in gene expression in *MYB.C2*-OE fruits (Fig 7A), with 3,382 genes upregulated and 2,549 genes downregulated compared to control fruits (S4B Fig; S6 Data). KEGG pathway analysis of the DEGs revealed significant enrichment in several key metabolic pathways, including flavonoid biosynthesis, amino sugar and nucleotide sugar metabolism, phenylpropanoid biosynthesis, glycerophospholipid metabolism, and nitrogen metabolism (Fig 7B; S7 Data), suggesting that PavMYB.C2 influences a broad range of metabolic processes, with a particular emphasis on flavonoid metabolism, which is of primary interest in this study. Further analysis identified a set of key genes involved in anthocyanin biosynthesis, including *CHI*, *CHS*, *F3H*, *F3'H*, *ANS*, *DFR*, and *UFGT*, which were significantly upregulated in *MYB.C2*-OE fruits (Fig 7C), indicating that MYB.C2 enhances the expression of these structural genes, thus promoting anthocyanin biosynthesis. Examination of the promoter regions of these genes revealed the presence of MYB binding sites (MBSs) (Fig 7D), supporting the hypothesis that PavMYB.C2 may regulate the transcription of these genes through direct binding to these conserved motifs.

## PavMYB.C2 binds to the *UFGT* promoter region through a specific binding site and directly activates its transcription

To investigate how PavMYB.C2 regulates anthocyanin biosynthesis at the transcriptional level, the complete *PavMYB.C2* gene was cloned into pGreenII 62SK vector to serve as the effector, and 1500 bp promoter regions upstream of *CHI, CHS, F3H, F3'H, ANS, DFR,* and *UFGT* genes were inserted into the PGL3 vector as reporter constructs, respectively. Renilla luciferase (REN), driven by the 35S promoter, was used as an internal control (Fig 8A). In *Nicotiana benthamiana* leaf protoplasts, co-transfection with the 35S:*MYB.C2* effector and the *proUFGT*:LUC reporter resulted in a significant increase in luciferase (LUC) activity, compared to the control effector (Fig 8B). No

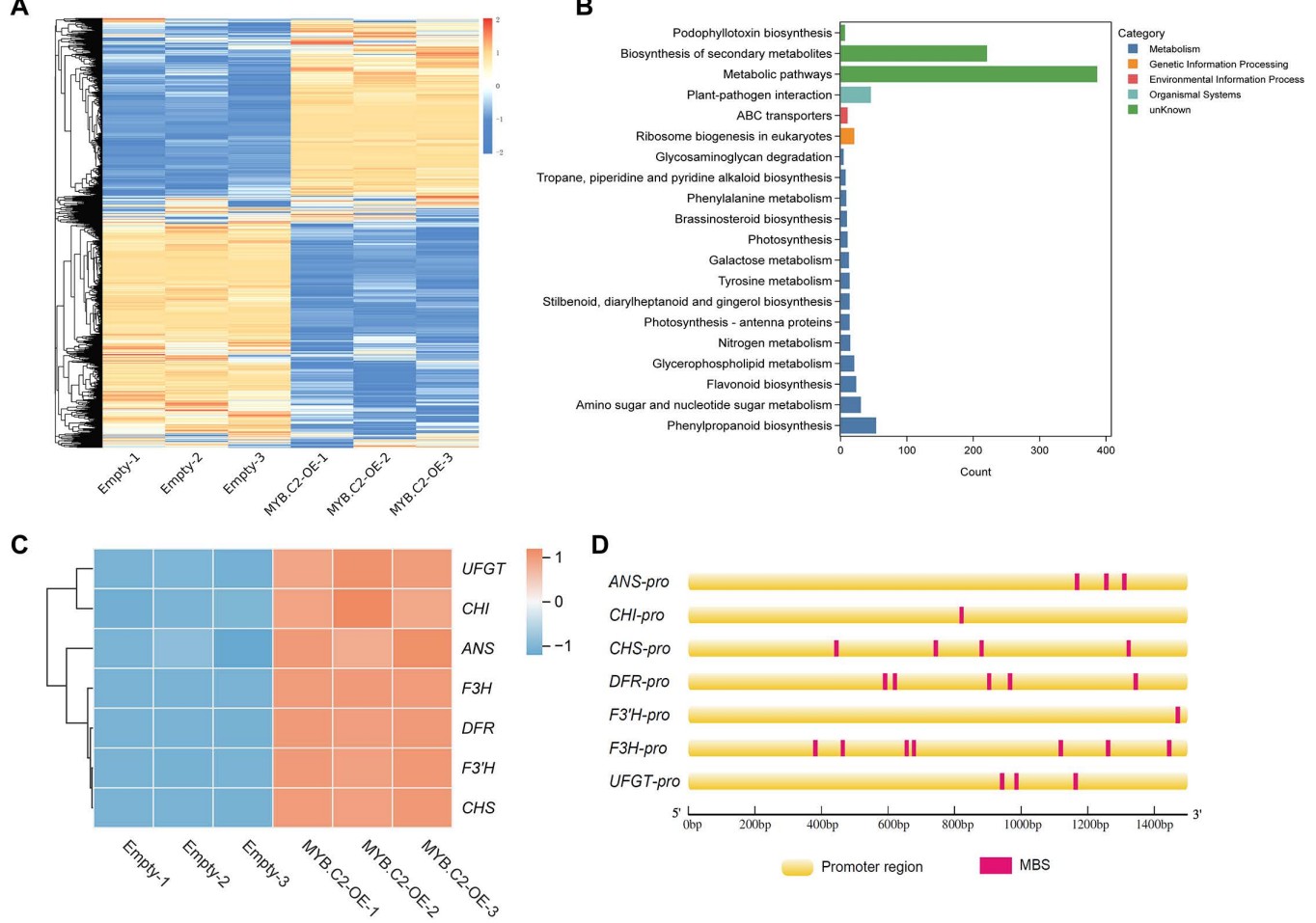

**Fig 7. Effect of PavMYB.C2 on the expression of key genes in anthocyanin synthesis. (A)** Heatmap displaying the differential expression of all genes following the overexpression of *PavMYB.C2* in sweet cherry fruit. **(B)** KEGG pathway analysis of differentially expressed genes (DEGs) in 'OE' fruits versus 'Empty' controls. **(C)** Heatmap showing the expression patterns of key genes involved in the anthocyanin biosynthetic pathway. **(D)** Identification of MYB-binding *cis*-acting elements in the promoter regions of key genes involved in anthocyanin biosynthesis. Yellow bars represent the promoter regions, while red rectangles denote the MYB binding sites (MBS).

such increase was observed with the promoters of the other six genes (Fig 8B), suggesting that PavMYB.C2 specifically activates *UFGT* transcription. Furthermore, we performed yeast one-hybrid (Y1H) assays. Yeast cells co-expressing *proUPGT*-AbAi and *MYB.C2*-AD constructs were able to grow on selective SD/-Leu/AbA plates, indicating the direct interaction between PavMYB.C2 protein and the *UFGT* promoter (Fig 8C). This result provides compelling evidence that PavMYB.C2 specifically binds to and regulates the *UFGT* gene *in vivo*, supporting its role in anthocyanin biosynthesis regulation.

To assess the binding specificity of PavMYB.C2 to the MYB-related *cis*-acting elements in the *UFGT* gene promoter, we performed electrophoretic mobility shift assays (EMSA) using three 50-bp probes, each containing one of the putative binding motifs: MYB-bingding site 1 (MBS1; CAACCA), MBS2 (AGAAACAA), and MBS3 (TGGTTA) (Fig 8D). The recombinant PavMYB.C2 protein was incubated with biotin-labeled probes, and competition assays were conducted using both unlabeled wild-type and mutated probes (Fig 8D). EMSA results demonstrated that PavMYB.C2 specifically bound to the probe containing MBS3, while no binding was observed with probes harboring MBS1 or MBS2 (Fig 8D). These findings

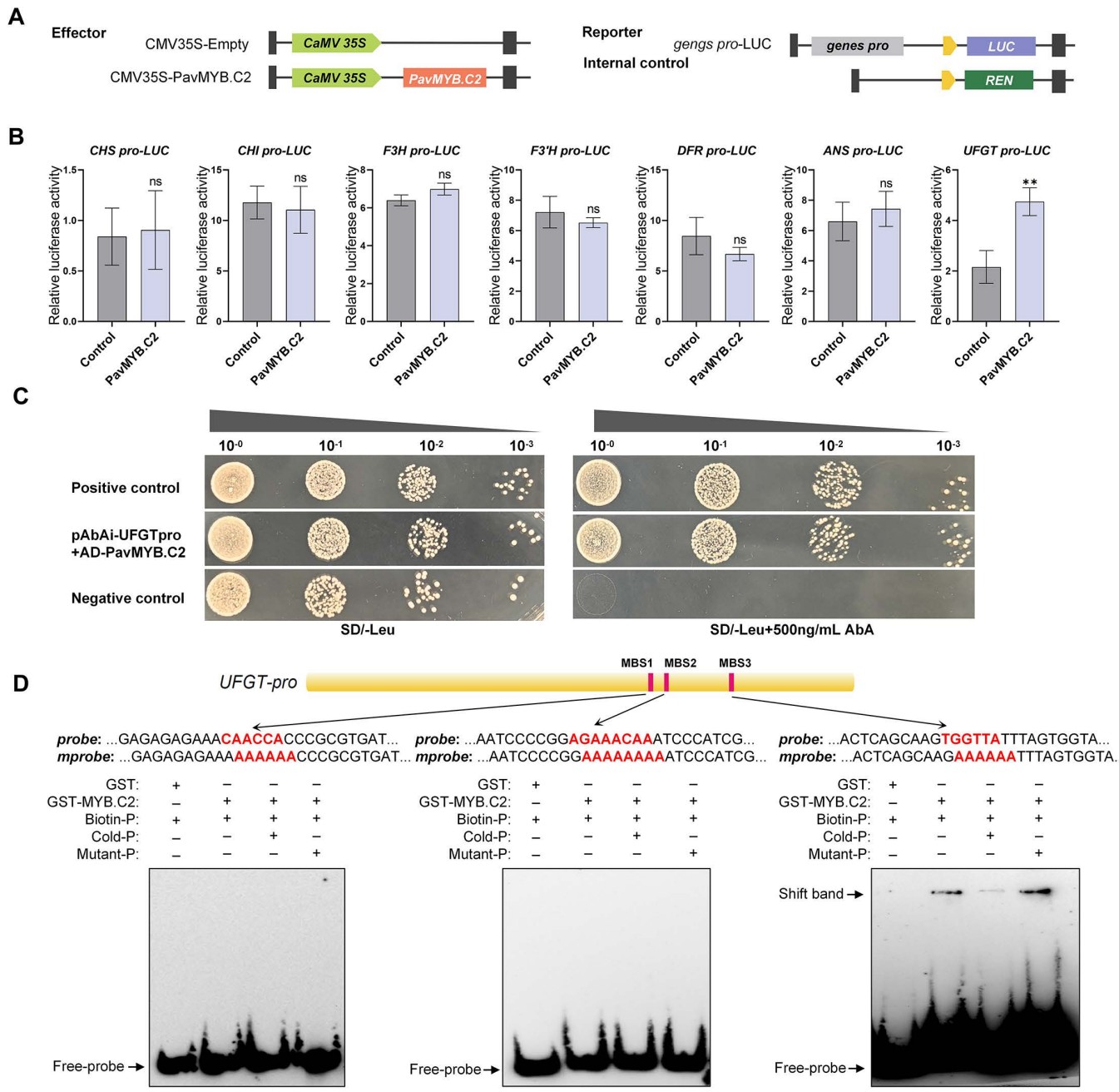

**Fig 8. PavMYB.C2 binds to the *UFGT* promoter via specific binding site and activates its transcription. (A)** Diagram of the effector and reporter constructs employed in the protoplast-based transient expression assay. **(B)** Effect of PavMYB.C2 on the transcriptional activation of genes involved in anthocyanin biosynthesis. Transcriptional activity is quantified by the ratio of firefly luciferase (LUC) to *Renilla luciferase* (REN) signals. **(C)** Yeast one-hybrid (Y1H) assays demonstrating the interaction between PavMYB.C2 and *UFGT* promoter region. The constructs were introduced into the Y1H bait strain, and growth was assessed on SD/-Leu medium containing 500 ng/mL aureobasidin A (AbA). The Y1HGold [pGADT7/pUFGT-AbAi] served as a negative control, while Y1HGold [pGADT7 Rec-p53/p53-AbAi] was used as a positive control. **(D)** Electrophoretic mobility shift assays (EMSAs) showing the direct binding of PavMYB.C2 to *UFGT* promoter *in vitro*. The promoter fragments were labeled with fluorescein. Cold-P represents the unlabeled promoter probe, while Mutant-P refers to the *UFGT* promoter probe with mutations in the MYB-binding sites (MBS1 and MBS3 mutated to AAAAAA; MBS2 mutated to AAAAAAAA). The symbols − and + denote the absence or presence, respectively. The left panel shows the binding of PavMYB.C2 to MBS1, the middle panel to MBS2, and the right panel to MBS3. Data are presented as means ± standard deviation (SD) (n = 3). Statistical significance was assessed using Student's *t*-tests. Asterisks denote significant differences compared to control at $p < 0.01$ (**); 'ns' indicates no significant difference compared to the control.

support the notion that PavMYB.C2 selectively recognizes and binds to the conserved MBS3 motif within the *UFGT* promoter region, thus enhancing the expression of *UFGT* and promoting anthocyanin biosynthesis.

## Activation of the *UFGT* by PavMYB.C2 is dependent on the S68 site in MYB domain

We generated a mutant version of *UFGT* promoter, referred to as mUFGTpro, in which the MBS3 binding site (TGGTTA) was substituted with "AAAAAA". Then we conducted a dual-luc assay in tobacco protoplasts. Our results showed that co-transfection of the 35S:PavMYB.C2 effector with the mUFGTpro:LUC reporter did not result in a significant change in LUC activity, when compared to the control effector (Fig 9A). This result further suggests that PavMYB.C2 regulates transcriptional expression of the *UFGT* through binding to the "TGGTTA" To further validate the role of UFGT in anthocyanin biosynthesis in cherry fruit, we transiently overexpressed and silenced *UFGT* gene in fruit, respectively (Fig 9B). And successful genetic transformation was confirmed by RT-qPCR analysis 15 days post-injection (Fig 9C). Phenotypic observations showed that *UFGT* overexpression (OE) fruits exhibited a deeper red coloration compared to controls, while *UFGT* silencing (VIGS) fruits appeared greener (Fig 9B), similar to the phenotype induced by PavMYB.C2 (Fig 6D). Furthermore, *UFGT*-OE fruits showed a significant increase in Cy3R and Cy3G levels, with Cy3G accumulation surpassing that of Cy3R (Fig 9D and 9E). In contrast, these anthocyanins were markedly reduced in *UFGT*-VIGS fruits (Fig 9D and 9E). These results confirm the essential role of UFGT in anthocyanin accumulation in sweet cherry fruit.

To further substantiate the potential uniqueness of PavMYB.C2 in regulating anthocyanin synthesis in sweet cherries, we performed a comparative analysis of the amino acid sequences of PavMYB.C2, AtCPC, AtTRY, AtETC1, and other CPC-related homologous proteins in *Arabidopsis*. Our analysis revealed two amino acid positions, D34 and M68, within the MYB structural domain that are highly conserved in *Arabidopsis* MYBs, but differ in PavMYB.C2, where D34 is substituted with T34 and M68 with S68 (Fig 9F). The [DE]Lx2[RK]x3Lx6Lx3R sequence, located within the MYB domain, is a critical region for the interaction of R3-MYBs with bHLH proteins [48,49]. Notably, D34 represents the first amino acid of this sequence (Fig 9F). Additionally, M68 has been shown to be essential for the function and intercellular movement of CPC proteins (corresponding residue M78 in CPC) [50]. Subsequently, we generated two mutated versions of PavMYB.C2: PavMYB.C2_D34, in which the amino acid T34 was substituted with D34, and PavMYB.C2_M68, in which the amino acid S68 was substituted with M68 (Fig 9G). These mutated constructs were then introduced into the pGreenII 62SK vector as effector constructs. Both constructs were co-transfected into tobacco leaf protoplasts along with the proUFGT:LUC reporter plasmid, respectively. Our results indicated that co-transfection of the proUFGT:LUC reporter plasmid with the 35S:PavMYB.C2 or PavMYB.C2_D34 plasmid resulted in a significant increase in relative LUC activity compared to the control (Fig 9H). However, when the 35S:PavMYB.C2_M68 plasmid was co-transfected, the transcriptional activation capacity was absent (Fig 9H). These findings suggest that the activation of *UFGT* by PavMYB.C2 is dependent on the S68 site, rather than the T34 site. Based on these data, we propose that the functional specificity of PavMYB.C2 in sweet cherry may be attributed to the variations at the S68 site, which play a critical role in its ability to regulate anthocyanin biosynthesis.

## Discussion

The genetic regulation of fruit color is governed by a complex network of genes that control pigment synthesis and accumulation. Identifying key genetic factors responsible for color traits can greatly enhance the efficiency of breeding programs focused on fruit color improvement [51]. In this study, we analyzed two sweet cherry cultivars exhibiting distinct color differences and constructed a regulatory network underlying anthocyanin metabolism through an integrated analysis of their metabolic and transcriptomic profiles. Our results pinpoint PavMYB.C2, a pivotal transcription factor, as a crucial regulator of anthocyanin accumulation, primarily through the modulation of the anthocyanin biosynthesis gene *UFGT* (Fig 10). This study not only advance our understanding of how total anthocyanin content influences fruit color but also emphasize the role of the relative abundance of individual anthocyanin compounds in determining fruit pigmentation.

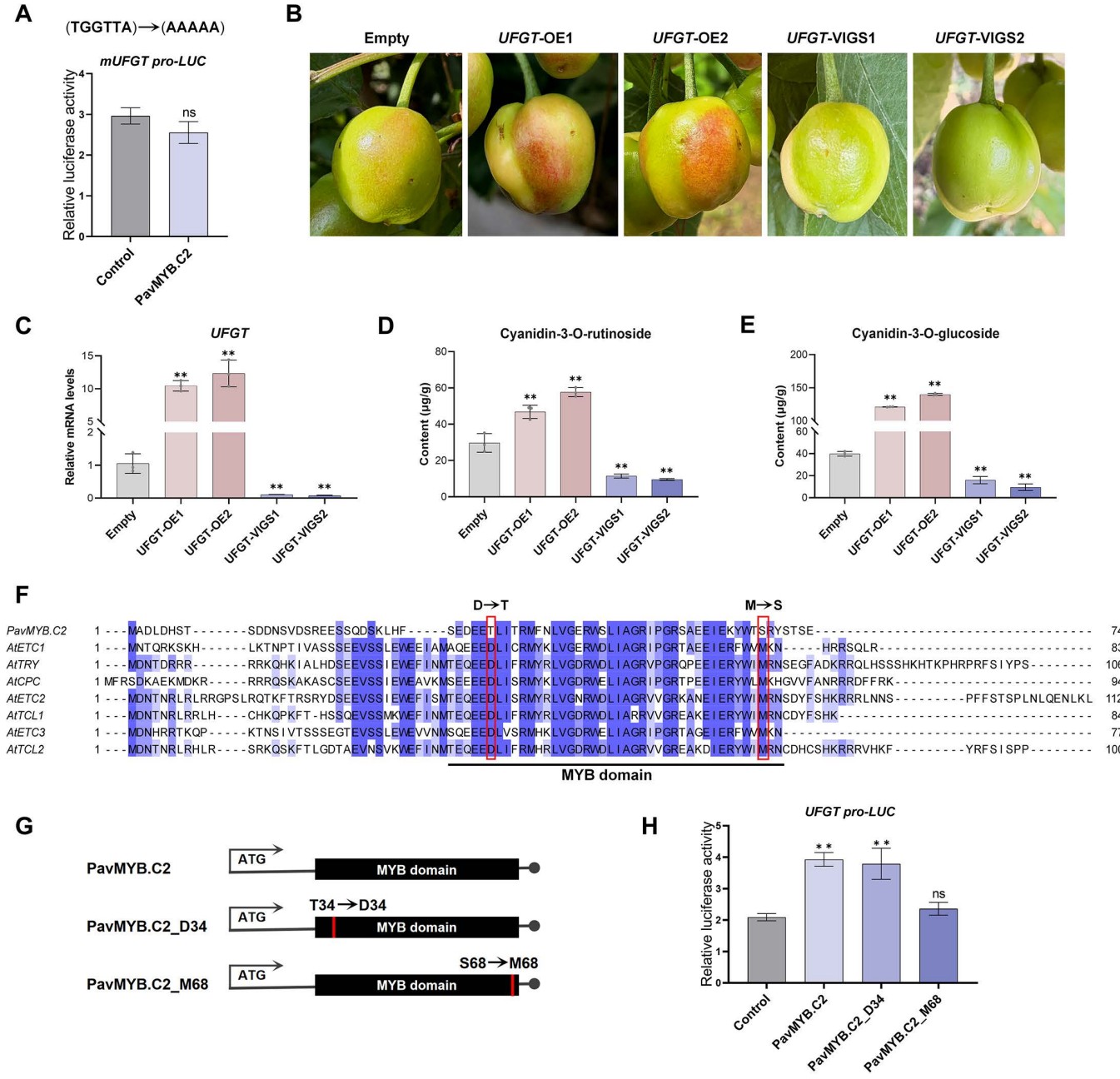

**Fig 9. Regulation of the *UFGT* by PavMYB.C2 is dependent on the S68 site of PavMYB.C2. (A)** Effect of PavMYB.C2 on transcriptional regulation of the *UFGT* promoter following mutation of MBS3 (TGGTTA mutated to AAAAA). Transcriptional activity is quantified by the ratio of LUC to REN signals. **(B)** Transient genetic transformation in sweet cherry fruit. 'OE' denotes transient overexpression of the *UFGT* gene, 'VIGS' refers to *UFGT* gene silencing, and 'Empty' indicates the injection of a corresponding empty vector. Images were captured 15 days post-injection. **(C)** Relative transcript levels of *UFGT* in 'Empty', 'OE', and 'VIGS' fruits 15 days after injection. **(D and E)** Quantification of two major anthocyanins in 'Empty', 'OE', and 'VIGS' fruits 15 days post-injection. D: Cyanidin-3-O-rutinoside (Cy3R); E: Cyanidin-3-O-glucoside (Cy3G). **(F)** Multiple protein sequence alignment of PavMYB.C2 and related MYBs from Arabidopsis. The red boxes highlight amino acids that are conserved in Arabidopsis MYBs but differ in PavMYB.C2. D: Aspartic acid; T: Threonine; M: Methionine; S: Serine. **(G)** Diagrams showing the PavMYB.C2 variants. PavMYB.C2_D34, a mutated version of PavMYB.C2 with the amino acid T34 substituted by D34; PavMYB.C2_M68, a mutated version of PavMYB.C2 with the amino acid S68 substituted by M68. **(H)** Effects of PavMYB.C2, PavMYB.C2_D34, and PavMYB.C2_M68 on the activation of *UFGT*. The empty pGreenII 62SK vector served as the negative control. Data are presented as means ± standard deviation (SD) (n = 3). Statistical significance was assessed using Student's *t*-tests. Asterisks denote significant differences compared to the control at $p < 0.05$ (*), $p < 0.01$ (**); 'ns' indicates no significant difference compared to the control.

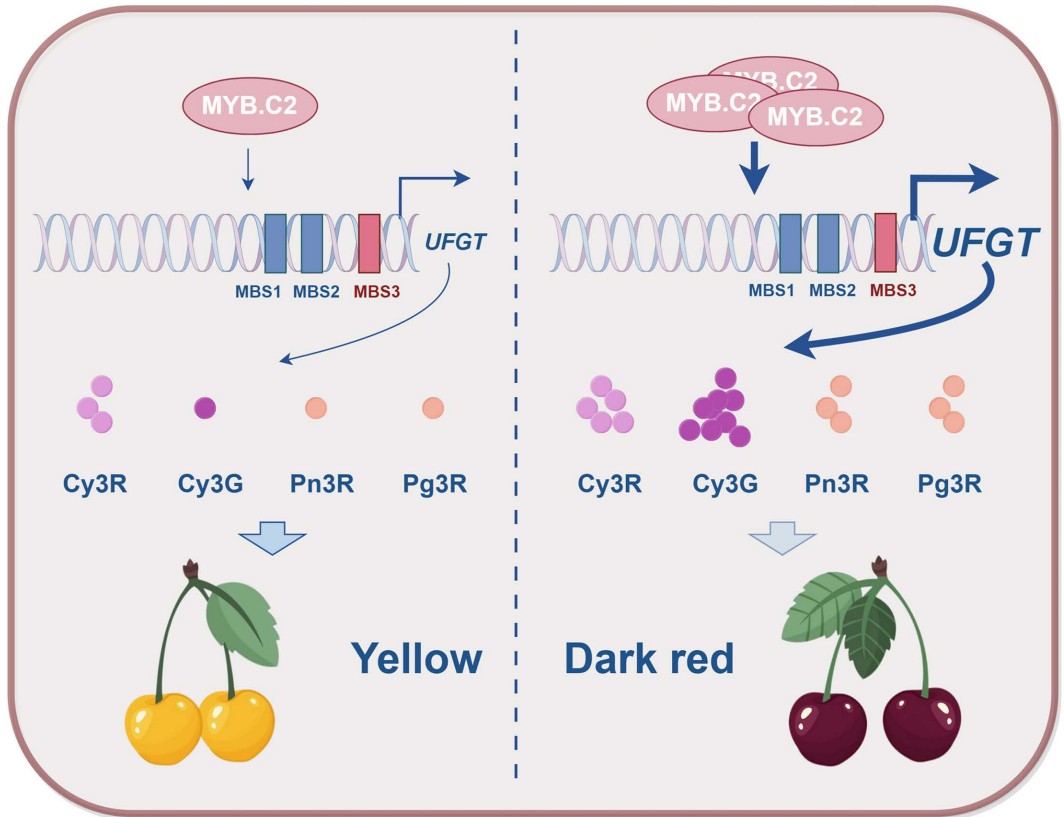

**Fig 10. A working model for the role of PavMYB.C2 in regulating fruit coloration.** Pav.MYB.C2 promotes the transcriptional expression of *UFGT*, a key gene in anthocyanin biosynthesis, by specifically binding to the MYB binding site 3 (MBS) within the *UFGT* promoter region. This interaction not only drives a significant increase in anthocyanin accumulation but also elevates the relative abundance of cyanidin-3-glucoside (Cy3G), a direct product of UFGT activity. The combined effects lead to enhanced fruit coloration, resulting in the development of deep red fruits with high anthocyanin content. The working model is drawn by Figdraw (www.figdraw.com, ID: ASOPOdaafa).

In this study we identified four predominant anthocyanins in sweet cherry fruits: Cy3R, Cy3G, Pn3R, and Pg3R (Fig 2). Through quantifying the relative abundance of these anthocyanins throughout different stages of fruit ripening, we observed a notable trend: the relative abundance of Cy3G is crucial for the development of the characteristic deep red color (Fig 2). Consistent with these findings, it was previously reported that Cy3R predominates in yellow-red cultivars such as 'Priusadebnaja' and 'Sue', while Cy3G is more abundant in red cultivars like 'Burlat' and 'Junska rana'—the latter being associated with a deeper red hue due to the higher concentration of Cy3G [52,53]. Additionally, our data revealed that MYB.C2 directly regulates *UFGT*, the key enzyme involved in the biosynthesis of Cy3G (Fig 8). Consequently, over-expression of *PavMYB.C2* led to a substantial increase in Cy3G content, while the levels of Cy3R, which is an indirect downstream product of UFGT, also increased, though to a relatively lesser extent. These findings underscore the importance of Cy3G in the pigmentation of sweet cherries and provide insights into the molecular mechanisms underlying the regulation of anthocyanin biosynthesis in fruit.

MYB proteins can form a MYB-bHLH-WDR (MBW) complex with bHLH and WDR (WD40-repeat) proteins, and activate the transcription of anthocyanin synthesis related genes [45]. Therefore, further research is needed to determine whether bHLH or WDR proteins can form an MBW complex with PavMYB.C2. Nonetheless, accumulating evidence suggests that MYB proteins can regulate anthocyanin biosynthesis independently of bHLH or WDR proteins, thereby influencing fruit

coloration [46,54–56]. For instance, the single gene *SlMYB75* has been shown to enhance anthocyanin accumulation in tomato fruit [46], and AcMYB266 promotes anthocyanin synthesis in pineapple by regulating the expression of key biosynthetic genes such as *UFGT* and *ANS* [54]. These findings are consistent with our studies that highlight the independent regulatory capacity of PavMYB.C2 in modulating anthocyanin levels in sweet cherry fruits.

The R3-type MYB protein, CPC, a class of transcriptional repressors involved in the anthocyanin biosynthesis, disrupts the MBW complex by competing with other MYB transcription factors for binding to bHLH proteins [50,57]. In this study, we identify PavMYB.C2 as a divergent member of the CPC-like MYB family that has evolved to function as an activator of anthocyanin synthesis in sweet cherry (Figs 6B and 9), in contrast to the canonical repressor role of AtCPC in *Arabidopsis*. Previous studies have demonstrated that mutations in conserved amino acid residues within MYB domains can lead to significant alterations in their functional properties [58–60]. Our analysis revealed that the amino acid position M68, located within the MYB domain, is highly conserved among *Arabidopsis* CPC proteins but differ in PavMYB.C2, where M68 is substituted with S68 (Fig 9F). We further demonstrated that the activation of *UFGT* by PavMYB.C2 is dependent on the S68 site (Fig 9H). This suggests that the functional divergence of PavMYB.C2 from CPC homologs may be attributed to this specific variation. As previously reported, the precise role of the S68 site in the function of PavMYB.C2 can be further elucidated in future research through techniques such as protein crystallization and structural determination [58].

RNA-seq data showed that PavMYB.C2 broadly influences the expression of multiple key genes involved in the anthocyanin biosynthetic pathway (Fig 7). However, both *in vivo* and *in vitro* experiments demonstrated that *UFGT* is the sole direct transcriptional target of PavMYB.C2. This observation may reflect the complexity of genetic regulatory networks, in which the overexpression of a single gene can induce a cascade of expression changes in other genes, even those not directly regulated by the gene in question [61,62]. Notably, while *UFGT* appears to be the primary direct target, we cannot exclude the possibility that cultivar-specific anthocyanin accumulation involves contributions from other genes, such as *ANS* and *DFR*. These genes may be regulated by additional transcription factors identified in the co-expression network's blue module, including PavWRK50, PavWRK47, PavNAC40, PavNAC14, PavERF113, and others (Fig 5B; S5 Data). We propose that PavMYB.C2 functions as a hub regulator, directly activating *UFGT* while indirectly influencing the broader anthocyanin pathway via synergistic or hierarchical interactions with other transcriptional components.

In summary, our study provided comprehensive metabolic and transcriptional profiles at five distinct ripening stages in two sweet cherry cultivars exhibiting notably different coloration. A detailed comparison of anthocyanin metabolic alterations during ripening highlighted that variations in both the concentrations and relative abundances of anthocyanins are key contributors to the observed differences in fruit coloration. Subsequent genetic and molecular analyses established the critical role of the PavMYB.C2 transcription factor in regulating the synthesis of Cyanidin-3-O-glucoside and overall anthocyanin production. Collectively, these findings offer valuable insights into the molecular mechanisms governing color development and anthocyanin metabolism in sweet cherries, providing a solid foundation for future efforts aimed at improving the quality of sweet cherry fruit.

## Materials and methods

### Plant materials and growth conditions

The sweet cherry cultivars 'Binghu' and 'Isabella' (*Prunus avium* L.), aged 9 years, were cultivated in a research orchard located in Wenchuan, Sichuan Province, China (N, 31.49477; E, 103.58453). The trees were characterized by uniform growth, freedom from pests and diseases, and were subjected to standardized management practices. Fruits at various stages of ripening were harvested, including the Br (color-breaking) stage, Br5 (5 days post-breaker), Br10, Br15, Br20, respectively. The fruits were de-seeded and cut into uniform blocks. For each stage, 20 fruits were pooled together to form a biological replicate, with three biological replicates in total for subsequent metabolic and transcriptomic analyses.

In addition, *Nicotiana benthamiana* (tobacco) plants were included in the study. These tobacco plants were grown under controlled conditions in a greenhouse, with a 16-hour light/8-hour dark photoperiod and a constant temperature of 25°C.

## Metabolome and transcriptome profiling

Metware Biotechnology Co., Ltd. (Wuhan, China) conducted the metabolite extraction and analysis. Anthocyanin-targeted metabolomic analysis was performed using LC-MS/MS, adhering to established protocols [63,64]. Sweet cherry fruit tissues were freeze-dried, powdered (30 Hz, 1.5 minutes), and extracted using 0.5 mL of a methanol/water/hydrochloric acid solution (500:500:1, v/v/v). The extract underwent a 5-minute vortex, followed by 5 minutes of ultrasonic treatment, and was then centrifuged at 12,000 g and 4°C for 3 minutes. The supernatants were filtered using a 0.22 μm membrane filter (Anpel) prior to LC-MS/MS analysis. Sample extracts were analyzed using an UPLC-ESI-MS/MS system, specifically the UPLC ExionLC AD and the Applied Biosystems 6500 Triple Quadrupole, both available at https://sciex.com.cn/.Anthocyanins were measured utilizing scheduled multiple reaction monitoring (MRM) mode. Data acquisition and metabolite quantification were conducted using Analyst 1.6.3 and Multiquant 3.0.3 software, respectively, both from Sciex.Significantly regulated metabolites (DMs) between groups (different cultivars at the same stage) were identified using absolute Log2FC (fold change), and Venn diagrams were created based on these DMs using online tools.

Wuhan Metware Biotechnology Co., Ltd. (Wuhan, China) conducted transcriptome sequencing, preparing cDNA libraries and sequencing them using the Illumina HiSeq 2000 system [65]. Raw sequencing reads were generated from the base-calling of the raw image data using CASAVA software. Paired-end reads were aligned to the sweet cherry reference genome using HISAT2, adhering to established protocols. Gene expression levels were quantified by calculating the number of reads mapped to each gene using featureCounts (v. 1.5.0-p3). FPKM values for each gene were calculated using gene length and the count of mapped reads. Differential expression was considered significant with a p-value below 0.05 and a fold change exceeding 1.5.Venn diagrams were generated based on different groups of DEGs on online websites (https://cloud.metware.cn/#/tools/). Differentially expressed genes (DEGs) were analyzed for KEGG pathway enrichment using the ClusterProfiler package in R.

## PCA analysis and clustering of metabolomes and transcriptomes

Gene and metabolite expression profiles at various ripening stages of two cultivars were analyzed using principal component analysis (PCA) and hierarchical clustering on the Metware Cloud platform (https://cloud.metware.cn). PCA reduced data dimensionality and highlighted key variation sources among samples, whereas hierarchical clustering grouped samples by similarities in gene and metabolite profiles. In addition, to gain a deeper understanding of the anthocyanin metabolic changes and gene expression in different cultivars during fruit ripening, heat maps were constructed using the ComplexHeatmap package in R software.

## Weighted correlation network analysis and metabolic network construction

We utilized weighted correlation network analysis (WGCNA) following the method as described previously [66]. Differential metabolites (DMs) and differentially expressed genes (DEGs) were selected and used to construct co-expression network modules using the WGCNA package in R. The co-expression modules were generated using the automatic network construction function (blockwiseModules) with default parameters. Subsequently, the most significant anthocyanin-related modules, exhibiting the highest correlation coefficients, were selected for further analysis. Transcriptional regulatory networks were constructed using Pearson correlation coefficients (PCC > 0.8) calculated between key differential metabolites, structural genes, and transcription factors. The network was visualized using Cytoscape (v.3.7.2, USA) [67].

### RNA isolation and RT-qPCR analysis

A 0.2 g aliquot of ultra-low temperature frozen sweet cherry fruit was placed into a mortar pre-chilled with liquid nitrogen. RNA was isolated from frozen samples using a kit from Chengdu BIOFIT Co., Ltd. (Chengdu, China). Reverse transcription and genomic DNA removal were performed according to the protocol outlined previously [68]. RNA integrity was evaluated by analyzing 3 µL of each sample using 1.0% agarose gel electrophoresis. Quantitative reverse transcription PCR (RT-qPCR) was performed using the Bio-Rad CFX96 Real-Time PCR System (BIO-RAD, USA) and 2×SP qPCR Mix (Bioground, Chongqing). Primer sequences for the expression analysis are provided in S1 Table. Each experiment included three biological replicates, with three technical replicates per biological sample.

### Determination of anthocyanin contents

The anthocyanin content was quantified as previously described [69], with minor modifications. In brief, sweet cherry samples were dissolved in 1% HCl-methanol, and the mixture was homogenized and extracted in the dark at 4°C for 12 hours. After extraction, the mixture was centrifuged, and the supernatant was filtered using a 0.22 µm organic membrane filter. A DAD detector and a Comatex C18 column (250 mm×4.6 mm, 5 µm) were used for high-performance liquid chromatography (HPLC) analysis. The mobile phases consisted of water, acetonitrile, and formic acid. The flow rate was set to 1 mL/min, with an injection volume of 10 µL. The column was kept at 30°C with a detection wavelength of 520 nm.

### Determination of ripening-related phenotypes

At each sampling stage, twenty fruits from each cultivar were collected and photographed for documentation. Fruit epidermal color at the equatorial plane was assessed with a Hunter Lab Mini Scan XE Plus colorimeter. Soluble solids content was measured using a handheld refractometer (PAL-1, Atago, Tokyo, Japan), while fruit firmness was evaluated with a texture analyzer (TA.XTC-18, BOSIN, Shanghai). Fruit soluble sugars were measured via the anthrone reagent method [70], and ascorbic acid levels were assessed using the 2,6-dichlorophenol indophenol titration method [71].

### Enzyme activity assay

Fruits from two cultivars were harvested at different stages, with three fruits pooled together to form one biological replicate. Three biological replicates were collected per treatment. The activities of key enzymes involved in anthocyanin biosynthesis, including anthocyanin key synthase, dihydroflavonol reductase (DFR), anthocyanin synthase (ANS), and UDP-flavonoid glucosyltransferase (UFGT), were measured in the fruit samples using enzyme activity assay kits (Solarbio, Beijing, China).

### Phylogenetic analysis

A BLAST analysis was conducted using the NCBI platform to identify potential MYB family protein sequences in *Arabidopsis thaliana* and *Prunus avium*. Phylogenetic trees were constructed using full-length amino acid sequences, MYB domain sequences, and full-length coding DNA sequences, respectively. These trees were generated via the maximum likelihood approach [72]. To assess the reliability of the phylogenetic tree, a bootstrap analysis with 1000 replicates [73] was conducted, with branches supported by less than 50% bootstrap value being excluded. The MYB proteins in *Prunus avium* were named based on their homology to *Arabidopsis* MYB proteins, as indicated by the phylogenetic tree. Sequence alignments were visualized with Jalview (version 2.11.3.2).

### Transient gene overexpression and virus-induced gene silencing in sweet cherry

Genetic materials were constructed following the method as previously described [74]. Specifically, to achieve overexpression of *PavMYB.C2* and *UFGT* in fruit, the coding sequence of *PavMYB.C2* and *UFGT* were cloned and inserted into

the pBI121 vector using homologous recombination. For virus-induced gene silencing (VIGS) of *PavMYB.C2* and *UFGT*, the partial coding sequence of *PavMYB.C2* and *UFGT* were amplified and ligated into the pTRV2 vector, also via homologous recombination. The constructs *PavMYB.C2*-pBI121, pTRV2-*PavMYB.C2*, *UFGT*-pBI121, pTRV2-*UFGT* and pTRV1 were then introduced into *Agrobacterium* strain GV3101. Agrobacterium cells were subsequently infiltrated into fruit at the equatorial region 20 days post-anthesis (DPA), with the empty vector serving as a control. Fifteen days post-infiltration, fruit samples were collected from the injection sites, quickly frozen in liquid nitrogen, and stored at -80°C. RT-qPCR was conducted to verify the genetic transformation by evaluating *PavMYB.C2* or *UFGT* expression levels.

### Subcellular localization

The subcellular localization of PavMYB.C2 was performed as described previously [75]. The full-length coding sequence of *PavMYB.C2* was amplified and cloned into a binary expression vector with a 35S promoter to create the 35S PavMYB.C2-GFP construct, facilitating GFP expression. The 35S PavMYB.C2-GFP construct and the empty 35S GFP vector were agroinfiltrated into *Nicotiana benthamiana* leaves, with co-expression of a nucleus-targeted mCherry marker to delineate nuclear regions. Following a 48-hour incubation at 22°C in the dark and a subsequent 24-hour exposure to light, the GFP signal localization was observed and documented using a Leica DM4 B fluorescence microscope.

### Overlapping PCR analysis

Overlapping PCR was employed to generate mutated sequences, including *mUFGT*, *PavMYB.C2_D34*, and *PavMYB.C2_M68*, following the procedures outlined by Pei et al. (2024) [76]. Mutagenic primers were designed to introduce specific mutations and used in PCR amplification with either a 5′ or 3′ primer (S1 Table). The resulting two PCR products, containing the desired mutations, were separated by agarose gel electrophoresis and subsequently purified using the Gel Purification Kit (TSP602-200, Tsingke). The purified fragments were then combined in a second round of PCR amplification to perform the overlapping PCR, assembling the full-length mutated sequence. The final amplified product was subsequently used for vector construction.

### Transient expression assays

To investigate the transcriptional regulation of anthocyanin biosynthetic genes by PavMYB.C2, the full-length *PavMYB.C2*, *PavMYB.C2_D34* and *PavMYB.C2_M68* were cloned into pGreenII 62SK vector to function as the effector, respectively, and 1500 bp promoter regions upstream of the *CHI, CHS, F3H, F3'H, ANS, DFR, UFGT* and *mUFGT* genes were each inserted into the PGL3 vector, generating corresponding reporter constructs. The Renilla luciferase (REN), driven by the 35S promoter, was used as an internal control. Protoplasts were isolated from *N. benthamiana* leaves according to previously described methods [77]. Protoplasts were co-transfected with different combinations of reporter and effector plasmids, along with the pRL Renilla luciferase control reporter plasmid, using PEG-mediated transfection as described by Deng et al. [68]. Luciferase and Renilla luciferase activities were quantified using a dual luciferase reporter assay kit (Promega E1910) following the manufacturer's guidelines.

### Yeast one-hybrid assay

The yeast one-hybrid (Y1H) assay was performed using the Matchmaker Gold Yeast One-Hybrid System (Clontech). The *UFGT* promoter bait fragment was amplified by PCR and inserted into the pAbAi vector. Linearized by BstBI digestion and transformed into Y1H Gold cells, and selected on synthetic dextrose medium agar plates without uracil. Full-length coding sequence of *PavMYB.C2* was cloned into the pGADT7 vector. The constructs were transformed into the Y1H bait strain and cultured on SD/-Leu medium supplemented with 500 ng/mL aureobasidin A (Clontech). Y1HGold [pGADT7/pUFGT-AbAi] was used as a negative control, and Y1HGold [pGADT7 Rec-p53/p53-AbAi] was used as a positive control.

## Electrophoretic mobility shift assays (EMSA)

Perform EMSA as previously described [68]. In brief, the coding sequence of *PavMYB.C2* was inserted into the pGEX-4T-1 vector to promote the expression of the recombinant PavMYB.C2-GST fusion protein in *Escherichia coli* BL21 cells. Recombinant proteins were purified using a GST-tag Protein Purification Kit (Beyotime). Use a 3'-biotin labeling kit (Thermo Fisher) to synthesize a biotin-labeled probe corresponding to the MBS *cis*-element of the target gene promoter. Competitive binding assays were performed using unlabeled wild-type and mutant probes. EMSA was performed using the EMSA/Gel-Shift Kit (Beyotime) combined with the LightShift chemiluminescent EMSA Kit (Thermo Fisher) according to the standard protocol.

## Statistical analysis

The results presented herein are represented as mean values accompanied by their respective standard deviations (SD), derived from three or more independent experimental replicates. Statistical analyses were conducted utilizing the student's *t*-test for pairwise comparisons.

## Supporting information

**S1 Fig. Comparison of different classes of anthocyanoside substances. (A)** Relative abundances of various anthocyanins in the fruits of two cultivars at different ripening stages. Different classes of metabolites are represented by distinct colors. **(B)** Colour of the four main anthocyanins Cyanidin-3-O-rutinoside (Cy3R), Cyanidin-3-O-glucoside (Cy3G), Peonidin-3-O-rutinoside (Pn3R), and Pelargonidin-3-O-rutinoside (Pg3R) at the same concentration (100ug/ml).
(PDF)

**S2 Fig. Analysis of DEGs between two cultivars under ripening stages. (A-C)** Volcano diagrams of DEGs in 'Isabella' and 'Binghu' fruits. A, Br10 stage; B, Br15 stage; C, Br20 stage. **(D-F)** KEGG pathway analysis of DEGs in 'Isabella' fruits compared to 'Binghu'. D, Br10 stage; E, Br15 stage; F, Br20 stage.
(PDF)

**S3 Fig. Phylogenetic analysis of the LOC110764247 (PavMYB.C2) homologs in *Arabidopsis thaliana* and *Prunus avium*. (A)** Phylogenetic tree constructed using amino acid sequences of MYB domains. **(B)** Phylogenetic tree constructed using full-length coding DNA sequence.
(PDF)

**S4 Fig. Transcriptome analysis of *MYB.C2*-OE and control fruits. (A)** PCA dendrogram of transcriptome data from *MYB.C2*-OE and control fruits. The oval represents the 95% confidence interval. **(B)** Volcano diagrams of DEGs in *MYB.C2*-OE fruits compared to control fruits.
(PDF)

**S1 Table. List of primers used in this study.**
(XLSX)

**S1 Data. The anthocyanin-related metabolome across five ripening stages of 'Binghu' (abbreviated as B) and 'Isabella' (abbreviated as I) sweet cherry fruits.**
(XLSX)

**S2 Data. Expression of all genes across five ripening stages of 'Binghu' (abbreviated as B) and 'Isabella' (abbreviated as I) sweet cherry fruits.**
(XLSX)

**S3 Data. List of differentially expressed genes (DEGs) in two cultivars across Br10, Br15 and Br20 stages.**
(XLSX)

**S4 Data. KEGG enrichment analysis of intersecting DEGs in two cultivars across Br10, Br15 and Br20 stages.**
(XLSX)

**S5 Data. Blue gene module related to anthocyanin synthesis identified by WGCNA.**
(XLSX)

**S6 Data. Expression levels of DEGs between the *MYB.C2*-OE and 'Empty' fruits at 15 days after injection.**
(XLSX)

**S7 Data. KEGG enrichment analysis of intersecting DEGs between *MYB.C2*-OE and 'Empty' fruits at 15 days after injection.**
(XLSX)

## Author contributions

**Conceptualization:** Yangang Pei, Ronggao Gong.

**Data curation:** Yangang Pei, Wanjia Tang, Ronggao Gong.

**Formal analysis:** Yangang Pei, Xiaowei Liu.

**Funding acquisition:** Yangang Pei, Ronggao Gong.

**Investigation:** Yangang Pei, Wanjia Tang, Yidi Huang, Hongfen Li, Hongxu Chen, Runmei He, Wenyi Niu, Quanyan Du, Yizhe Chu.

**Methodology:** Yangang Pei, Heng Deng, Mingchun Liu, Ronggao Gong.

**Project administration:** Yangang Pei, Hongxu Chen, Ronggao Gong.

**Resources:** Heng Deng, Mingchun Liu, Ronggao Gong.

**Supervision:** Mingchun Liu, Ronggao Gong.

**Validation:** Mingchun Liu, Ronggao Gong.

**Visualization:** Yangang Pei, Wanjia Tang.

**Writing – original draft:** Yangang Pei.

**Writing – review & editing:** Mingchun Liu, Ronggao Gong.

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
