## [Decision Letter · Decision Letter 0]

PGENETICS-D-24-01543

The PavMYB.C2-UFGT module contributes to fruit coloration via modulating anthocyanin biosynthesis in sweet cherry

PLOS Genetics

Dear Dr. Gong,

Thank you for submitting your manuscript to PLOS Genetics. After careful consideration, we believe it meets PLOS Genetics's publication criteria, pending revision and re-review. Therefore, we invite you to submit a revised version of the manuscript that addresses the points raised during the review process.

Please submit your revised manuscript within 60 days Apr 26 2025 11:59PM. If you will need more time than this to complete your revisions, please reply to this message or contact the journal office at plosgenetics@plos.org. Please include the following items when submitting your revised manuscript:

We look forward to receiving your revised manuscript.

Kind regards,

Tomo Kawashima

Academic Editor

PLOS Genetics

Aimée Dudley

Editor-in-Chief

PLOS Genetics

Anne Goriely

Editor-in-Chief

PLOS Genetics

**Additional Editor Comments (if provided):**

All reviewers appreciate your work, and there is strong consensus that the presented study is of broad interest. Please carefully review all the comments. As Reviewer 1 pointed out, please clarify the potential uniqueness of the MYB TF, likely with additional tests. Please submit your raw omics data to any public depository. Please do not hesitate to contact me if you need futher clarifications.

**Journal Requirements:**

**Reviewers' comments:**

Reviewer's Responses to Questions

**Comments to the Authors:**

Reviewer #1: In this study, authors compared two cultivars of sweet cherry based on differential accumulation of anthocyanins leading to variations in fruit coloration. They conducted extensive metabolic analysis to study the total anthocyanin content, and the relative abundance of individual anthocyanin species are critical contributors of the color variation in these cultivars. Bae on the transcriptomic and metabolic data they Are integrating transcriptomic data, they identified a MYB transcription factor PavMYB.C2, which is exhibits homology with MYB transcription factors like CPC (CAPRICE) proteins in Arabidopsis. The author performed transactivation assay to test the possible regulatory role of PavMYB.C2 in controlling anthocyanin biosynthesis. They found that PavMYB.C2 can bind and promote UFGT promoter’s transcriptional activity.

Based on the pattern of transcriptional regulation anthocyanin biosynthesis is conventionally divided into two stages, early and late stage. Genes belonging to the early stage of the anthocyanin biosynthesis is predominantly controlled positively by R2R3MYB type of transcription factors, while the genes in later stage (including UFGT) in controlled by the MBW complex, comprising R2R3MYB-bHLH-WD40 transcription factor. CARPRICE (CPC) CPC belongs to the R3-MYB subgroup, meaning it has only a single MYB DNA-binding domain. R2R3-MYBs, which typically activate gene expression either single or as part of the MBW complex. Unlike R2R3 MYB, CPC acts as a repressor of certain MYB-regulated pathways, including anthocyanin biosynthesis, trichome, and root hair development. CPC lacks transcriptional activation domain and functions by competing with R2R3-MYBs for binding to bHLH proteins, preventing the formation of the MBW complex needed for activating late anthocyanin biosynthetic genes like UFGT.

From the result provided in the manuscript, it is assumed that PavMYB.C2 belongs to the R3-MYB subgroup. Does PavMYB.C2 contain one repeat or two repeats? This is a unique case where a R3MYB factor can directly bind DNA and positively regulate gene expression, especially UFGT, a gene belonging to the very late stage of anthocyanin biosynthesis. A wealth of knowledge available regarding the transcriptional regulation of anthocyanin in different plants. These are not discussed in the manuscript. I would suggest test the transcriptional activity of the PavMYB.C2 on the UFGT promoter where “TGGTAA” site is mutated. Does

As mentioned earlier, DNA binding in MYB transcription factors primarily depends on the R2 and R3 MYB repeats, which together recognize and bind to specific promoter sequences. A Single-repeat MYBs (R3-MYBs). Such as CPC, TRY, ETC1/2 only have the R3 domain, which alone is not sufficient for stable DNA binding. Additional experiments are required to test the proposed model. Despite species-specific variations, the core regulatory network involving MYB, bHLH, and WD40 are highly conserved across plants. I would suggest that the author should also test whether CPC/TRY/ETC can activate the UFGT promoter in a similar fashion. Also in vitro binding capacity should be tested. If the data presented in this manuscript are accurate, they have the potential to significantly enhance our understanding of the mechanisms by which MYB transcription factors regulate gene expression. These findings may provide new insights that could refine or even reshape existing models of MYB-mediated transcriptional regulation.

Reviewer #2: Anthocyanins are vital secondary metabolites responsible for fruit coloration and health benefits. Thus, it is the key step to explore the genetic mechanisms regulating anthocyanin biosynthesis in fruits. It is the case reported in this manuscript. Using cherry fruit as materials, the authors identified a PavMYB.C2-UFGT module, and provide solid evidence to support this module contributing to fruit coloration via modulating anthocyanin biosynthesis.

These findings in this study provide a potential important candidate for molecular breeding to improve fruit coloration and health benefits. However, there are some obscure things to be solved before final acceptance. Below are my suggestions to improve this manuscript.

1. Checking the legend of Figure 5, it is not right for B-G. In addition, it is better to express the unit of enzyme activity based on mg protein, instead of fresh weight.

2. It is recommended to verify the function of UFGT in cherry fruit as PavMYB.C2.

3. In Figure 6C, it is better to include a nucleus marker or DAPI.

Reviewer #3: The study employs a comprehensive multi-omics approach, integrating metabolomics, transcriptomics, and functional validation, to explore the regulatory mechanisms underlying anthocyanin biosynthesis in sweet cherry. The identification of PavMYB.C2 as a pivotal transcriptional regulator for UFGT and anthocyanin accumulation represents a novel advancement in this field. Functional validation through transient overexpression and silencing experiments convincingly highlights the role of PavMYB.C2 in modulating anthocyanin composition, particularly cyanidin-3-glucoside (Cy3G), thereby enhancing our understanding of MYB transcription factors in anthocyanin regulation. However, several issues need to be addressed before publication.

Major Comments

The authors demonstrate that PavMYB.C2 is a key transcriptional regulator of UFGT, which results in differing anthocyanin accumulation between the 'Binghu' and 'Isabella' cultivars. However, it is noted that 'Binghu' exhibits a less pronounced increase in the expression levels of DFR, ANS, and UFGT, whereas these genes are markedly upregulated in 'Isabella'. This indicates that apart from UFGT, the varying expression levels of DFR and ANS also contribute to the differential anthocyanin accumulation between the two cultivars. Please discuss potential contributors other than PavMYB.C2.

Given that PavMYB.C2 regulates the synthesis of cyanidin-3-O-glucoside and overall anthocyanin production, what could explain its differential expression between the two cultivars despite their similar genetic backgrounds?

Experimental Timing for Agrobacterium Infiltration:The manuscript describes agrobacterium infiltration at 20 days post-anthesis (DPA) and sample collection fifteen days post-infiltration. Why were these specific time points chosen?

EMSA Data and Binding Motifs:The EMSA data (Figure 8D) show binding to MBS3 but not MBS1/MBS2. However, the rationale for selecting these motifs and their conservation across species is unclear. A phylogenetic analysis of the UFGT promoter region or comparison with known MYB-binding motifs in other plants (if available) would strengthen this claim.

Data Availability:While the manuscript states that“all relevant data are within the manuscript and Supporting Information,” raw transcriptomic and metabolomic datasets should be deposited in public repositories (e.g., NCBI GEO) to ensure reproducibility.

Some references related to anthocyanins are outdated and need to be updated with more recent literature.

**Have all data underlying the figures and results presented in the manuscript been provided?**

Reviewer #1: Yes

Reviewer #2: None

Reviewer #3: None

PLOS authors have the option to publish the peer review history of their article (what does this mean? ). If published, this will include your full peer review and any attached files.

**Do you want your identity to be public for this peer review?** For information about this choice, including consent withdrawal, please see our Privacy Policy .

Reviewer #1: No

Reviewer #2: No

Reviewer #3: **Yes: ** Biao Jin

**Figure resubmission:**
---

## [Decision Letter · Decision Letter 1]

PGENETICS-D-24-01543R1

The PavMYB.C2-UFGT module contributes to fruit coloration via modulating anthocyanin biosynthesis in sweet cherry

PLOS Genetics

Dear Dr. Gong,

Thank you for submitting your manuscript to PLOS Genetics. We invite you to submit a revised version of the manuscript (minor revision) that addresses the points raised during the review process. Please see the additional editor comments (below) for a description of what I feel is most important to address.

Please submit your revised manuscript within 30 days Jun 20 2025 11:59PM. If you will need more time than this to complete your revisions, please reply to this message or contact the journal office at plosgenetics@plos.org. Please include the following items when submitting your revised manuscript:

We look forward to receiving your revised manuscript.

Kind regards,

Tomo Kawashima

Academic Editor

PLOS Genetics

Aimée Dudley

Editor-in-Chief

PLOS Genetics

**Additional Editor Comments (if provided):**

Reviewer 1 has raised two important points. While the authors have provided some explanations, these could be clarified and emphasized further. Please consider the following suggestions as minor revisions:

Based on the presented data (phylogenetic tree as well as Myb domain AA conservation), PavMYB.C2 appears to have diverged significantly from both the AtCPC and PavMYB.C1 supgroup. Moreover, unlike AtCPC, which functions as a repressor, PavMYB.C2 acts as an activator. These functional and evolutionary differences should be more clearly highlighted and emphasized in the manuscript to avoid confusion. This clarification also addresses Reviewer 1’s second concern.

Additionally, please clarify whether Figure 6B is based on the full-length amino acid sequence or only the MYB domain. Have you also analyzed sequence homology at the DNA level? More detailed data and a clearer description of how homology was assessed (AA vs. DNA vs. domain-based) would strengthen the manuscript and aid reader understanding.

**Journal Requirements:**

**Reviewers' comments:**

Reviewer's Responses to Questions

**Comments to the Authors:**

Reviewer #1: The authors performed a number of experiments to demonstrate that PavMYB.C2 positively regulates the expression of UFGT in sweet cherry, thereby influencing color development. However, I still have some concerns and questions:

It is unclear why the authors chose to test the effect of Arabidopsis transcription factors, such as CPC and ETC, on sweet cherry UFGT. Did they not identify any orthologs of these factors in sweet cherry? If suitable orthologs were absent, the relevance of testing Arabidopsis transcription factors in this context is questionable and may not yield biologically meaningful insights.

I am also curious whether the authors investigated the potential role of PavMYB.C2 in forming or inhibiting the MBW complex, which is well-known to regulate anthocyanin biosynthesis in sweet cherry. I strongly recommend performing such experiments to better understand the mechanistic function of PavMYB.C2 within the regulatory network.

Reviewer #2: In this revised version of the manuscript, the authors have added additional experiments and answered the questions which I addressed, it now is acceptable for publication.

Reviewer #3: The authors have addressed all my questions and concerns, and the revised manuscript has shown significant improvement.

**Have all data underlying the figures and results presented in the manuscript been provided?**

Reviewer #1: Yes

Reviewer #2: None

Reviewer #3: None

PLOS authors have the option to publish the peer review history of their article (what does this mean? ). If published, this will include your full peer review and any attached files.

**Do you want your identity to be public for this peer review?** For information about this choice, including consent withdrawal, please see our Privacy Policy .

Reviewer #1: No

Reviewer #2: No

Reviewer #3: No

**Figure resubmission:**
---

## [Editor Report · Decision Letter 2]

Dear Dr Gong,

We are pleased to inform you that your manuscript entitled "The PavMYB.C2-UFGT module contributes to fruit coloration via modulating anthocyanin biosynthesis in sweet cherry" has been editorially accepted for publication in PLOS Genetics. Congratulations!

Yours sincerely,

Tomo Kawashima

Academic Editor

PLOS Genetics

Aimée Dudley

Editor-in-Chief

PLOS Genetics

Aimée Dudley

Editor-in-Chief

PLOS Genetics

Anne Goriely

Editor-in-Chief

PLOS Genetics

Comments from the reviewers (if applicable):

**Data Deposition**

http://datadryad.org/submit?journalID=pgenetics&manu=PGENETICS-D-24-01543R2

**Press Queries**

---

## [Editor Report · Acceptance letter]

PGENETICS-D-24-01543R2

The PavMYB.C2-UFGT module contributes to fruit coloration via modulating anthocyanin biosynthesis in sweet cherry

Dear Dr Gong,

We are pleased to inform you that your manuscript entitled "The PavMYB.C2-UFGT module contributes to fruit coloration via modulating anthocyanin biosynthesis in sweet cherry" has been formally accepted for publication in PLOS Genetics! Your manuscript is now with our production department and you will be notified of the publication date in due course.

With kind regards,

Zsofia Freund

PLOS Genetics

On behalf of:
